# PCP-MAE: Learning to Predict Centers for Point Masked Autoencoders

**Xiangdong Zhang**[*]**, Shaofeng Zhang**[*]**, Junchi Yan**[‡]
Dept. of CSE & School of AI & MoE Key Lab of AI, Shanghai Jiao Tong University
{zhangxiangdong, sherrylone, yanjunchi}@sjtu.edu.cn

## Abstract

Masked autoencoder has been widely explored in point cloud self-supervised learning, whereby the point cloud is generally divided into visible and masked parts. These methods typically include an encoder accepting visible patches (normalized) and corresponding patch centers (position) as input, with the decoder accepting the output of the encoder and the centers (position) of the masked parts to reconstruct each point in the masked patches. Then, the pre-trained encoders are used for downstream tasks. In this paper, we show a motivating empirical result that when directly feeding the centers of masked patches to the decoder without information from the encoder, it still reconstructs well. In other words, the centers of patches are important and the reconstruction objective does not necessarily rely on representations of the encoder, thus preventing the encoder from learning semantic representations. Based on this key observation, we propose a simple yet effective method, *i.e.*, learning to **P**redict **C**enters for **P**oint **M**asked **A**uto**E**ncoders (**PCP-MAE**) which guides the model to learn to predict the significant centers and use the predicted centers to replace the directly provided centers. Specifically, we propose a Predicting Center Module (PCM) that shares parameters with the original encoder with extra cross-attention to predict centers. Our method is of high pre-training efficiency compared to other alternatives and achieves great improvement over Point-MAE, particularly surpassing it by **5.50% on OBJ-BG, 6.03% on OBJ-ONLY, and 5.17% on PB-T50-RS** for 3D object classification on the ScanObjectNN dataset. The code is available at https://github.com/aHapBean/PCP-MAE .

## 1  Introduction

Point clouds are a widely used representation of 3-D objects, offering a rich expression of their geometric information. This versatility has led to their broad adoption across various application scenarios, including autonomous driving [28], robotics [10, 30], and the metaverse [32]. In the early developing stage of point cloud understanding, there are lots of related work proposed to enhance the ability of networks for understanding point cloud [13, 18, 25, 26, 39] where the networks typically need fully-supervised training from scratch. However, point cloud data are difficult to annotate compared to the data in 2-D vision and NLP owing to the complexity of the toughness of discriminating them. It leads to a phenomenon called *data desert* [9] in 3-D which refrains the development of these fully-supervised methods. Recently, numerous self-supervised learning (SSL) methods [47, 24, 9, 27] were proposed for point cloud understanding to alleviate the negative effect brought by *data desert* because self-supervised learning methods can utilize unlabeled data effectively by performing designed pretext task (*e.g.*, reconstruction-based or contrastive-based) for pre-training and the learned semantic representation benefits the performance of downstream tasks.

---

[*]Equal contribution. [‡]Corresponding author. This work was supported by Shanghai Municipal Science and Technology Major Project under Grant 2021SHZDZX0102.

Masked autoencoders (MAEs) [16] represent a prominent and robust framework within self-supervised learning (SSL), which demonstrates remarkable scalability and superior performance. The architecture of MAE is distinctly asymmetric, featuring an encoder and a decoder. The typical process involves dividing an image into patches, which are then selectively masked prior to encoding. The encoder only accepts visible patches as input along with their positional embedding, *i.e.* embedding of patch indices. The decoder receives both the latent representations of the visible patches and the tokens representing the masked patches, along with their corresponding positional embeddings. Following MAE in 2-D, Point-MAE [24] and its variants [48, 51, 14, 27] are proposed, which divide point cloud into patches where points in patches undergo normalization, achieved by subtracting the coordinates of corresponding centers and apply the mask-reconstruction paradigm in 3-D domain.

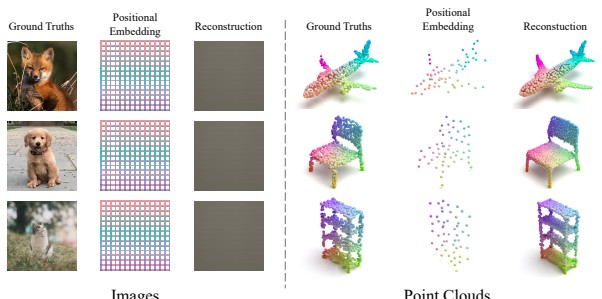

However, different from the indices of image patches (the positional embeddings in 2-D are fixed for all input images [16]), position embedding in point cloud is calculated using the coordinates of the centers belonging to patches, where the coordinates of the centers usually contain rich geometric and semantic information of point clouds. Thus we ask: **Should the centers (*i.e.*, positional embedding) for masked patches be directly given when performing masked reconstruction in the point clouds field just like the way in 2-D vision?** We answer this question with an interesting phenomenon observed through experimental results in Fig. 1: We first set the mask ratio to 100% in Point-MAE for pre-training on ShapeNet [3], *i.e.*, the encoder is removed to verify whether the point cloud can be reconstructed by only feeding the masked positional embeddings of the center points to the decoder after training. Intriguingly, the reconstruction results are pretty well, which are shown in Fig. 1. This is irreproducible for 2-D MAE since when masking 100% patches and only providing the positional embedding of the masked patches (*i.e.*, 2-D indices of patches), 2-D MAE cannot identify what the images are (as all the images with the same resolution share the same positional embeddings).

Figure 1: Illustrations of MAE reconstruction results for 2-D MAE and Point-MAE when masking ratio equals to 100%.

The gap between 2-D and point cloud data reveals that the coordinates of the centers (*i.e.* positional embeddings) are essential in the point cloud field, meaning that the decoder can even abandon the output of the encoder and still reconstruct well. Thus, the **reconstructing objective may make the encoder unable to learn semantic features**, and we argue that the answer to the previous question is: The centers for masked patches **can not** be directly provided and should be utilized in a smarter and more proper way. To this end, we propose a method, which learns to **P**redict **C**enters for **P**oint **M**asked **A**uto**E**ncoders, termed as **PCP-MAE**. While existing MAE-based methods [24, 9, 27, 48, 51] overlook the importance of centers during pre-training and directly leak the masked centers for the decoder, our PCP-MAE learns more semantic representation by performing point cloud reconstruction under the auxiliary of learning to predict centers, using predicted centers replace real masked centers.

The overview of the proposed PCP-MAE is illustrated in Fig. 2. Specifically, we propose a custom pre-training task that forces the encoder to not only learn representations for visible patches but also predict the centers of the masked ones. It is achieved by feeding both visible and masked patches to the encoder. Visible patches (with centers) perform self-attention in each block as in Point-MAE, while masked patches (without centers) leverage a cross-attention mechanism simultaneously to acquire center information from themselves and the visible ones. Additionally, a dedicated objective is introduced to minimize the difference between the predicted masked centers and the ground truth values (positional embeddings). Furthermore, the predicted positional embeddings of the masked centers are used to replace the real positional embeddings, which will be fed into the decoder. This allows the network to not only encode visible centers effectively but also learn the inter-relationships between visible and masked centers. Essentially, the encoder learns to infer the missing information, leading to a more robust representation and improved performance in downstream tasks achieved through fine-tuning. The main contributions include:

**1) Center-aware objective makes the pre-training trivial.** We identify the difference between 2-D (image) and 3-D (point cloud), *i.e.*, positional embedding (PE) in 2-D represents indices of patches,

Table 1: Comparisons between our PCP-MAE and existing Single/Cross-Modal MAE-based methods in terms of method features, pre-training efficiency, and performance on standard SSL benchmarks.

| Methods | Masked Centers Leakage | Single/Cross-Modal | Pre-trained Model Needed | Pre-training efficiency | | | Performance | |
|---|---|---|---|---|---|---|---|---|
| | | | | # Params | GFLOPS | Time (h) | ScanObjectNN | ModelNet40 |
| Point-MAE [24] | ✔ | Single | ✗ | 29.0 (baseline) | 2.0 (baseline) | 8.7 (baseline) | 85.18 (baseline) | 93.2 (baseline) |
| Point-M2AE [50] | ✔ | Single | ✗ | 15.3 (0.53×) | 3.2 (1.6×) | 19.1 (2.2×) | 86.43 (↑1.25) | 93.4 (↑0.2) |
| Point-FEMAE [48] | ✔ | Single | ✗ | 41.5 (1.43×) | 4.4 (2.2×) | 32.2 (3.7×) | 90.22 (↑5.04) | 94.0 (↑0.8) |
| ACT [9] | ✔ | Cross | ✔ | 135.5 (4.67×) | 23.0 (13.5×) | 34.8 (4.0×) | 88.21 (↑3.03) | 93.2 (↑0.0) |
| I2P-MAE [51] | ✔ | Cross | ✔ | 74.9 (2.58×) | 14.6 (7.3×) | 42.6 (4.9×) | 90.11 (↑4.93) | 93.7 (↑0.5) |
| ReCon [27] | ✔ | Cross | ✔ | 140.9 (4.85×) | 18.2 (9.1×) | 44.4 (5.1×) | 90.63 (↑5.45) | 94.1 (↑0.9) |
| PCP-MAE | ✗ | Single | ✗ | 29.5 (1.02×) | 2.9 (1.5×) | 12.3 (1.4×) | 90.35 (↑5.17) | 94.0 (↑0.8) |

while PE in point cloud provides coordinates of patch centers, where the indices usually provide much less information than coordinates. As shown in Fig. 1, point clouds are well reconstructed by the decoder under 100% masking ratio, indicating that the decoder does not necessarily rely on the representations learned by the encoder. This phenomenon does not hold for 2-D MAE, suggesting that existing MAE-based methods in point cloud adopt a trivial reconstruction pre-training task.

**2) A new center-unaware pretraining method.** We propose PCP-MAE, a framework that not only performs point cloud reconstruction but also avoids direct leakage of semantically rich centers by guiding the network to learn to predict them, which is much more challenging than previous reconstruction-only-based methods, and forcing the model to learn more semantic representations. Besides, the Predicting Center Module (PCM) is introduced to predict the positional embedding of the centers, which shares parameters with the encoder to prevent an increase in parameter count.

**3) Significant improvement over Point-MAE and high pre-training efficiency.** Compared to other MAE-based methods, our method only introduces a few parameters increase and affordable extra pre-training time cost, remaining highly efficient for pre-training as shown in Tab. 1. Notably, extensive experiments demonstrate the effectiveness and the efficiency of our PCP-MAE over other MAE-based methods by improving Point-MAE. Specifically, our method surpasses baseline Point-MAE by large margins of 5.50%, 6.03%, and 5.17% across three ScanObjectNN variants, respectively.

## 2 Related Work

**Self-supervised learning for Point Cloud.** Inspired by the recent successes of Self-Supervised Learning (SSL) in NLP and 2-D vision, an increasing amount of research has been conducted to explore the potential and strengths of SSL in the realm of 3-D vision [24, 47, 41, 1]. Existing 3-D SSL methods can be mainly divided into two categories: contrastive learning [41, 1, 17, 29, 19] and generative learning [47, 24, 50, 48, 51]. PointContrast [41] pioneers the contrastive learning in 3-D by borrowing the idea of contrastive learning in 2-D and performing point-level invariant mapping learning on two transformed views of the given point cloud. CrossPoint [1] advances this approach by engaging in intra-modal learning while also introducing an auxiliary cross-modal contrastive objective that facilitates the learning of transferable 3-D point cloud representations through 3D-2D correspondence. Generative learning in 3-D typically trains autoencoders to learn semantic latent representation during pre-training with a pre-designed task, $i.e.$, reconstructing the original input. Inspired by BERT, Point-BERT [47] firstly devises a Masked Point Modeling (MPM) task to pre-train point cloud Transformers auxiliary by a point cloud Tokenizer with a discrete Variational AutoEncoder (dVAE). Masked Autoencoders (MAE) [16], which mask random patches from the input image and reconstruct the missing patches at the pixel level, have demonstrated substantial potential and established a significant milestone in the field of SSL.

**MAE-based methods for point cloud.** Following the masked-reconstruction paradigm established by MAE in vision, Point-MAE [24] is proposed, which pioneers designing a neat and efficient architecture entirely built upon standard Transformer blocks for point cloud understanding. Motivated by the success of Point-MAE, numerous works based on it are proposed [50, 9, 27, 14, 51, 48], showing notable improvement compared to Point-MAE. On the basis of Point-MAE, Point-M2AE [50] tailors the encoder and decoder into pyramid architectures, which allows for progressive modeling of spatial geometries, facilitating the capture of both intricate details and overarching semantics of 3D shapes. ACT [9], as an MAE-based method, acquires knowledge from other modalities by adopting cross-modal auto-encoders as teachers. ReCon [27] harmoniously combines the generative framework (MAE-based) and contrastive framework to share the merits of them. Joint-MAE [14] enables better cross-modal interaction by constructing two hierarchical 2-D-3-D embedding modules. I2P-MAE [51] introduce a 2-D guided masking strategy and 2-D semantic reconstruction apart from

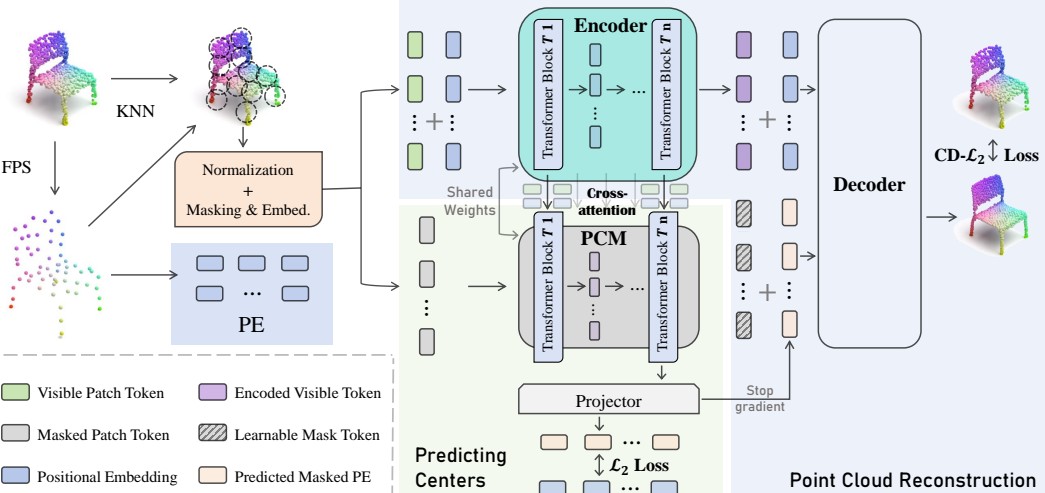

Figure 2: Overview of the proposed PCP-MAE. After patch division, the centers and normalized patches are divided into visible and masked parts, with center coordinates embedded into positional embedding (PE) and patches embedded into tokens (embeddings). The encoder accepts visible tokens and PE as input, performing self-attention. Simultaneously, the weight-shared PCM (Predicting Center Module) performs cross-attention (masked tokens as query and visible along with masked tokens as key and value) to acquire knowledge to predict the positional embeddings of the masked patches. CD-$\mathcal{L}_2$ refers to the $l_2$ Chamfer Distance loss function [11].

point cloud reconstruction. Point-FEMAE [48] extends global random masking to two branches consisting of a global branch and a local branch to capture latent semantic features, with extra modules Local Enhancement Modules introduced. We find almost all existing MAE-based methods focus on structural improvement, employing the same reconstruction objective as Point-MAE. However, as shown in Fig. 1, we claim again that this reconstruction objective is not suitable to directly adopt from 2-D to point clouds.

## 3 PCP-MAE

### 3.1 Patches generation and masking

**Patches generation.** Patch-based learning paradigm [47, 24, 27] has been proven to be more effective than directly consuming the whole point cloud [25, 26] empirically. We adopt the patch-based paradigm and divide the input point cloud into point patches via Farthest Point Sampling (FPS) and K-Nearest Neighborhood (KNN) algorithm following previous work [24]. Specifically, given a point cloud with $p$ points $\mathbf{X} \in \mathbb{R}^{p \times 3}$, FPS is first applied to sample $n$ centers $\mathbf{C}$ from $p$ points. Then, we adopt KNN to select $k$ nearest points of corresponding centers to consist $n$ point patches $\mathbf{P}$:

$$\mathbf{C} = \text{FPS}(\mathbf{X}), \ \mathbf{P} = \text{KNN}(\mathbf{X}, \mathbf{C}), \quad \mathbf{C} \in \mathbb{R}^{n \times 3}, \ \mathbf{P} \in \mathbb{R}^{n \times k \times 3} \tag{1}$$

Coordinates of points in the point patch are normalized with the corresponding center, thus enabling better separation between patches and centers.

**Masking.** With a pre-defined masking ratio $m$, we apply global random patch masking to point patches. The masked patches are denoted as $\mathbf{P}_m \in \mathbb{R}^{\lfloor mn \rfloor \times k \times 3}$ and the visible patches are denoted as $\mathbf{P}_v \in \mathbb{R}^{\lceil (1-m)n \rceil \times k \times 3}$, where $\lfloor \cdot \rfloor$ and $\lceil \cdot \rceil$ represent the floor function and the ceiling function, respectively. For simplicity, we will omit these two symbols in the subsequent text. The corresponding centers are denoted as $\mathbf{C}_m \in \mathbb{R}^{mn \times 3}$ and $\mathbf{C}_v \in \mathbb{R}^{(1-m)n \times 3}$.

**Embedding.** $\mathbf{P}_v$ is embedded via a lightweight PointNet [25] to yield encoded tokens $\mathbf{E}_v$ following Point-BERT [47] which can be expressed as follow, where $D$ is the hidden dimension of the networks.

$$\mathbf{E}_v = \text{PointNet}(\mathbf{P}_v), \quad \mathbf{E}_v \in \mathbb{R}^{n \times D} \tag{2}$$

We calculate fixed positional embedding for each center using the popular form [16] as detailed in Eq. (3), which we call sin-cos positional embedding (sin-cos PE). For each triplet of coordinates $(x, y, z)$ for the center:

$$\mathbf{PE}_x = \left[ \sin\left( \frac{x}{e^{2 \times \frac{1}{D/6}}} \right), \cos\left( \frac{x}{e^{2 \times \frac{1}{D/6}}} \right), \sin\left( \frac{x}{e^{2 \times \frac{2}{D/6}}} \right), \right.$$
$$\left. \cos\left( \frac{x}{e^{2 \times \frac{2}{D/6}}} \right), \dots, \sin\left( \frac{x}{e^2} \right), \cos\left( \frac{x}{e^2} \right) \right] \tag{3}$$

where $\mathbf{PE_x} \in \mathbb{R}^{1 \times D/3}$. Then we concatenate $\mathbf{PE_x}$, $\mathbf{PE_y}$ and $\mathbf{PE_z}$ to obtain $\mathbf{PE} \in \mathbb{R}^{1 \times D}$. Then feed the obtained sin-cos PE into a learnable MLP called PEM (Positional Embedding Module) to yield the final PE, which is different from the operation in Point-MAE [24]. On the one hand, the sin-cos PE enables more sparse positional information compared to $3\text{-}dimension$ coordinate and thus is more suitable as the predicting target of our proposed task just as illustrated in Sec. 4.3. On the other hand, applying the sin-cos positional embedding to both visible and masked patches can better align the usage of known visible PE with the predicted masked PE. The process can be expressed as:

$$\mathbf{PE}_v^{sin-cos} = \text{sin-cos PE}(\mathbf{C}_v), \quad \mathbf{PE}_v^{sin-cos} \in \mathbb{R}^{(1-m)n \times D} \tag{4}$$
$$\mathbf{PE}_m^{sin-cos} = \text{sin-cos PE}(\mathbf{C}_m), \quad \mathbf{PE}_m^{sin-cos} \in \mathbb{R}^{mn \times D} \tag{5}$$
$$\mathbf{PE}_v = \text{PEM}(\mathbf{PE}_v^{sin-cos}), \quad \mathbf{PE}_v \in \mathbb{R}^{(1-m)n \times D} \tag{6}$$
$$\mathbf{PE}_m = \text{PEM}(\mathbf{PE}_m^{sin-cos}), \quad \mathbf{PE}_m \in \mathbb{R}^{mn \times D} \tag{7}$$

## 3.2 Encoder

We employ an encoder that consists of standard Transformer blocks. Given the encoded visible tokens $\mathbf{E}_v$, masked tokens $\mathbf{E}_m$ and visible positional embeddings $\mathbf{PE}_v$, the encoder performs self-attention to encode $\mathbf{E}_v$ while performing cross-attention utilizing $\mathbf{E}_m$ as input to acquire information of masked centers from itself and the visible representations. Note the self-attention and cross-attention share the parameters of Transformer blocks to avoid increasing parameters.

**Visible tokens encoding.** The latent representation of the visible parts can be obtained by inputting $\mathbf{E}_v$, $\mathbf{PE}_v$ to the encoder which utilizes self-attention mechanisms. The process can be formulated as:

$$\mathbf{T}_v = \text{Encoder}(\mathbf{E}_v, \mathbf{PE}_v) \tag{8}$$

Specifically, the self-attention in $i$-th Transformer block of the encoder can be expressed as:

$$\mathbf{T}_i = \text{Attn}(\mathbf{Q}_v, \mathbf{K}_v, \mathbf{V}_v) = \text{SoftMax}\left( \frac{\mathbf{Q}_v \mathbf{K}_v}{\sqrt{D}} \right) \mathbf{V}_v \tag{9}$$

where $\mathbf{Q}_v = \mathbf{T}_{i-1}\mathbf{W_Q}$, $\mathbf{K}_v = \mathbf{T}_{i-1}\mathbf{W_K}$ and $\mathbf{V}_v = \mathbf{T}_{i-1}\mathbf{W_V}$ and the $\mathbf{W_Q}$, $\mathbf{W_K}$, $\mathbf{W_V}$ are learnable parameters. Note $\mathbf{T}_0 = \mathbf{E}_v + \mathbf{PE}_v$.

**Modules for learning to predict centers** *i.e.*, **PCM.** The PCM shares parameters with the encoder and accepts only masked patches $E_m$ as input. Apart from the shared parameters of Transformer blocks, additional MLP for projecting latent predicted $\mathbf{PE}$ to masked center embeddings$\mathbf{PE}_m^{pred}$ is introduced into PCM. Based on the encoded masked patches, the PCM aims to recover the centers of masked patches by acquiring knowledge from both the masked and visible representations. This can be straightforwardly understood in a form where local patterns (without telling where they are) and visible global patterns are given, and the task is acquiring positions of local patterns from the visible patterns based on the known local patterns. The cross-attention mechanisms are performed:

$$\mathbf{PE}_m^{pred} = \text{PCM}(\mathbf{E}_m, \mathbf{E}_v, \mathbf{PE}_v) \tag{10}$$

Specifically, for the $i$-th Transformer block in PCM, the cross-attention can be formulated as:

$$\mathbf{PE}_m^i = \text{Attn}(\mathbf{Q}_m, \mathbf{K}_{v+m}, \mathbf{V}_{v+m}) = \text{SoftMax}\left( \frac{\mathbf{Q}_m \mathbf{K}_{v+m}}{\sqrt{D}} \right) \mathbf{V}_{v+m} \tag{11}$$

where $\mathbf{Q}_m = \mathbf{PE}_m^{i-1}\mathbf{W_Q}$, $\mathbf{K}_m = \mathbf{PE}_m^{i-1}\mathbf{W_K}$ and $\mathbf{V}_m = \mathbf{PE}_m^{i-1}\mathbf{W_V}$ and the $\mathbf{W_Q}$, $\mathbf{W_K}$ are shared parameters. We concatenate $\mathbf{Q}_v$ and $\mathbf{Q}_m$ to obtain $\mathbf{Q}_{v+m}$ and it's also the case for $\mathbf{K}_{v+m}$ and $\mathbf{V}_{v+m}$. Note $\mathbf{PE}_m^0$ is the input of PCM, *i.e.*, $\mathbf{PE}_m^0 = \mathbf{E}_m$.

## 3.3 Decoder

The decoder also consists of Transformer blocks and is used for point cloud masked reconstruction. The decoder in existing MAE-based methods such as Point-MAE [24], ReCon [27], Point-FEMAE [48] typically take $\mathbf{T}_v$, $\mathbf{PE}_v$, $\mathbf{PE}_m$ as input along with learnable mask tokens $[\mathbf{M}] \in \mathbb{R}^{mn \times D}$. It can be expressed as $H_m = \mathrm{Decoder}([\mathbf{M}], \mathbf{PE}_m, \mathbf{T}_v, \mathbf{PE}_v)$ where the centers for the masked patches are directly provided. However, the centers are extremely important in point cloud as shown in Fig. 1. Therefore, instead of doing that, we use the $\mathbf{PE}_m^{pred}$ to replace the $\mathbf{PE}_m$ which means we use what the network learns but doesn't leak the ground truth of positional embeddings to the decoder. The process can be written as:

$$\mathbf{H}_m = \mathrm{Decoder}([\mathbf{M}], \mathrm{sg}(\mathbf{PE}_m^{pred}), \mathbf{T}_v, \mathbf{PE}_v) \tag{12}$$

where $\mathrm{sg}(\cdot)$ means the stop-gradient operation and $\mathbf{H}_m \in \mathbb{R}^{mn \times D}$. Stop-gradient is applied to prevent the decoder from finding reconstruction shortcuts by back-propagating through the $\mathbf{PE}_m^{pred}$ that modifies the weights of PCM, as illustrated in the ablation study Sec. 4.3. Finally, following Point-MAE, we adopt a projection head consisting of one MLP layer to predict the coordinates of normalized masked points in the input point cloud:

$$\mathbf{P}_{pred} = \mathrm{Reshape}(\mathrm{Linear}(\mathbf{H}_m)), \quad \mathbf{P}_{pred} \in \mathbb{R}^{mn \times k \times 3} \tag{13}$$

## 3.4 Objective function

The objective function $\mathcal{L}$ consists of two parts, *i.e.*, recovering the centers of masked patches and the points in every masked patch. The $\mathcal{L}$ can be written as $\mathcal{L} = \eta \mathcal{L}_{PC} + \mathcal{L}_{Recon}$. $\eta$ is a scaling factor.

**Learn to predict centers.** We compute the loss for predicting centers using the $l_2$ loss function:

$$\mathcal{L}_{PC} = \frac{1}{mnD} \|\mathbf{PE}_m^{pred} - \mathbf{PE}_m\|_2^2 \tag{14}$$

The information of centers is removed when generating point cloud patches through normalization, thus posing no possibility for PCM to find shortcuts to recover the masked centers $\mathbf{PE}_m$ based on $\mathbf{E}_m$, $\mathbf{E}_v$ and $\mathbf{PE}_v$. And recovering the $\mathbf{PE}_m$ is non-trivial because it needs the PCM to encode masked patches well and demands the visible representation $T_v$ possessing not only intra-information but also inter-information between itself and the unseen masked ones. In other words, minimizing the $\mathcal{L}_{PC}$ enables the visible representation $T_v$ to contain enough information to infer the distribution of masked centers which is infeasible in the original Point-MAE.

**Point cloud reconstruction.** The reconstruction loss $\mathcal{L}_{Recon}$ is computed using the $l_2$ Chamfer Distance loss function [11], written as:

$$\mathcal{L}_{Recon} = \frac{1}{|\mathbf{P}_{pred}|} \sum_{a \in \mathbf{P}_{pred}} \min_{b \in \mathbf{P}} \|a - b\|_2^2 + \frac{1}{|\mathbf{P}|} \sum_{b \in \mathbf{P}} \min_{a \in \mathbf{P}_{pred}} \|a - b\|_2^2 \tag{15}$$

# 4 Experiments

We conduct experiments including object classification, few-shot learning and segmentation to demonstrate the superior performance of our method over Point-MAE. With slightly longer pre-training time than Point-MAE, our PCP-MAE achieves much higher performance than it and even surpasses other more complex methods, achieving state-of-the-art in some tasks, *e.g.*, ScanObjectNN classification (OBJ-BG; OBJ-ONLY) and ModelNet40 few-shot (5-way, 10/20-shot; 10-way, 20-shot). Ablation studies are conducted to illustrate the properties of our proposed PCP-MAE.

## 4.1 Pre-training setups

**Model architecture.** The backbone of our pre-trained PCP-MAE consists of standard Transformer blocks where the encoder has 12 Transformer blocks and the decoder has 4, aligned with Point-MAE [24]. The hidden dimension of Transformer blocks is 384 and the number of heads is 6.

**Pre-training dataset.** PCP-MAE is pre-trained on ShapeNet [3] which consists of about 51,300 clean 3-D models, covering 55 common object categories.

**Experiment details.** For each input point cloud, we first apply scale and translate operations, followed by rotation for pre-training data augmentation. After sampling 1024 points via Farthest

Table 2: Classification accuracy (%) on ScanObjectNN and ModelNet40. The parameters of inference models #P (M) are reported. Three variants are evaluated on ScanObjectNN and the accuracies obtained on ModelNet40 are reported for both without and with voting.

| Methods | References | #P | ScanObjectNN | | | | ModelNet40 | | |
| | | | Input | OBJ_BG | OBJ_ONLY | PB_T50_RS | Input | w/o Vote | w/ Vote |
|---|---|---|---|---|---|---|---|---|---|
| *Supervised Learning Only* | | | | | | | | | |
| PointNet [25] | CVPR 2017 | 3.5 | 1k Points | 73.3 | 79.2 | 68.0 | 1k Points | 89.2 | - |
| PointNet++ [26] | NeruIPS 2017 | 1.5 | 1k Points | 82.3 | 84.3 | 77.9 | 1k Points | 90.7 | - |
| DGCNN [37] | TOG 2019 | 1.8 | 1k Points | 82.8 | 86.2 | 78.1 | 1k Points | 92.9 | - |
| SimpleView [12] | ICML 2021 | - | 6 Images | - | - | 80.5±0.3 | 6 Images | 93.9 | - |
| MVTN [15] | ICCV 2021 | 11.2 | 20 Images | 92.6 | 92.3 | 82.8 | 12 Images | 93.8 | - |
| PointMLP [23] | ICLR 2022 | 12.6 | 1k Points | - | - | 85.4±0.3 | 1k Points | 94.5 | - |
| SFR [49] | ICASSP 2023 | - | 20 Images | - | - | 87.8 | 20 Images | 93.9 | - |
| P2P-HorNet [36] | NeruIPS 2022 | 195.8 | 40 Images | - | - | 89.3 | 40 Images | 94.0 | - |
| *with Single-Modal Self-Supervised Learning* | | | | | | | | | |
| Point-BERT [47] | CVPR 2022 | 22.1 | 1k Points | 87.43 | 88.12 | 83.07 | 1k Points | 92.7 | 93.2 |
| MaskPoint [20] | ECCV 2022 | - | 2k Points | 89.30 | 88.10 | 84.30 | 1k Points | - | 93.8 |
| Point-MAE [24] | ECCV 2022 | 22.1 | 2k Points | 90.02 | 88.29 | 85.18 | 1k Points | 93.2 | 93.8 |
| Point-M2AE [50] | NeurIPS 2022 | 15.3 | 2k Points | 91.22 | 88.81 | 86.43 | 1k Points | 93.4 | 94.0 |
| PointGPT [4] | NeruIPS 2023 | 19.5 | 2k Points | 91.60 | 90.00 | 86.90 | 1k Points | - | 94.0 |
| Point-FEMAE [48] | AAAI 2024 | 27.4 | 2k Points | 95.18 | 93.29 | 90.22 | 1k Points | 94.0 | **94.5** |
| **PCP-MAE** | - | 22.1 | 2k Points | **95.52** | **94.32** | 90.35 | 1k Points | 94.0 | 94.2 |
| *Improve (over Point-MAE)* | - | - | - | +5.50 | +6.03 | +5.17 | - | +0.8 | +0.4 |
| *with Cross-Modal Self-Supervised Learning* | | | | | | | | | |
| ACT [9] | ICLR 2023 | 22.1 | 2k Points | 93.29 | 91.91 | 88.21 | 1k Points | 93.2 | 93.7 |
| Joint-MAE [14] | IJCAI 2023 | - | 2k Points | 90.94 | 88.86 | 86.07 | 1k Points | - | 94.0 |
| I2P-MAE [51] | CVPR 2023 | 15.3 | 2k Points | 94.14 | 91.57 | 90.11 | 1k Points | 93.7 | 94.1 |
| TAP [38] | ICCV 2023 | 22.1 | 2k Points | 90.36 | 89.50 | 85.67 | | - | - |
| ReCon [27] | ICML 2023 | 43.6 | 2k Points | 95.18 | 93.63 | **90.63** | 1k Points | **94.1** | **94.5** |

Point Sampling (FPS) from the input point cloud, it is divided into 64 point patches, each containing 32 points, selected using FPS and K-Nearest Neighbors (KNN). The PCP-MAE is pre-trained for 300 epochs using an AdamW optimizer [22] with a batch size of 128. The initial learning rate is set at 0.0005, with a weight decay of 0.05. A cosine learning rate decay scheduler [21] is utilized. Check the detailed experimental settings in Appendix A.

## 4.2 Fine-tuning on downstream tasks

**3-D object classification on a real-world dataset.** We showcase the transferability of our models by evaluating them on a real-world 3D object dataset. Specifically, we transfer our pre-trained model to ScanObjectNN [33] for classification, a dataset renowned for its classification complexity, encompassing around 15,000 real-world objects spanning 15 diverse categories. Our experimental evaluations encompass three variants: OBJ-BG, OBJ-ONLY, and PB-T50-RS. Rotation is applied for data augmentation during training following [27, 48]. The Tab. 2 presents the results, showing the superior performance of our PCP-MAE, which obtains great performance not only over Point-MAE but also over other more complex MAE-based methods and achieves state-of-the-art on OBJ-BG and OBJ-ONLY. Specifically, PCP-MAE outperforms Point-MAE by **5.50%**, **6.03%**, **5.17%** on three variants and performs competitively or better than the leading cross-modal method ReCon [27].

**3-D object classification on a dataset with clean objects.** We assess the efficacy of our pre-trained model for object classification on the ModelNet40 dataset [40]. ModelNet40 comprises 12,311 meticulously crafted 3-D CAD models, representing 40 distinct object categories. Standard random scaling and random translation are applied for data augmentation during training. Notably, the input point clouds exclusively contain coordinate information, without supplementary normal information. The results are presented in Tab. 2 including results with and without voting. Improvements in PCP-MAE over Point-MAE can be observed, 0.8% and 0.4% for w/o and w/ voting respectively.

**Few-shot learning.** We conduct few-shot learning experiments on the ModelNet40 dataset, following established protocols [24, 31]. The experiments are structured as "w-way, s-shot" and consist of four components, where $w \in \{5, 10\}$, representing the number of randomly selected classes, and $s \in \{10, 20\}$, indicating the number of randomly selected samples for each $w$. Each component undergoes 10 independent trials. The reported results include the mean accuracy and standard deviation, as detailed in Tab. 3. Results show that with limited downstream finetuning data, our PCP-MAE exhibits outstanding generalization ability among existing single-modal and cross-modal methods, achieving SOTA in 5-way, 10-shot; 5-way, 20shot; 10-way, 20-shot experiments. Specifically, 1.1%, 1.3%, 0.9%, 0.9% performance gains over Point-MAE on four settings are obtained.

Table 3: Few-shot classification results on ModelNet40. We perform ten separate trials for each experimental setting and the mean accuracy (%) and standard deviation are reported.

| Methods | 5-way | | 10-way | |
|---|---|---|---|---|
| | 10-shot | 20-shot | 10-shot | 20-shot |
| *Supervised Learning Only* | | | | |
| PointNet [25] | 52.0±3.8 | 57.8±4.9 | 46.6±4.3 | 35.2±4.8 |
| DGCNN [37] | 31.6±2.8 | 40.8±4.6 | 19.9±2.1 | 16.9±1.5 |
| OcCo [35] | 90.6±2.8 | 92.5±1.9 | 82.9±1.3 | 86.5±2.2 |
| *with Single-Modal Self-Supervised Representation Learning* | | | | |
| Point-BERT [47] | 94.6±3.1 | 96.3±2.7 | 91.0±5.4 | 92.7±5.1 |
| MaskPoint [20] | 95.0±3.7 | 97.2±1.7 | 91.4±4.0 | 93.4±3.5 |
| Point-MAE [24] | 96.3±2.5 | 97.8±1.8 | 92.6±4.1 | 95.0±3.0 |
| Point-M2AE [50] | 96.8±1.8 | 98.3±1.4 | 92.3±4.5 | 95.0±3.0 |
| PointGPT [4] | 96.8±2.0 | 98.6±1.1 | 92.6±4.6 | 95.2±3.4 |
| Point-FEMAE [48] | 97.2±1.9 | 98.6±1.3 | **94.0±3.3** | 95.8±2.8 |
| PCP-MAE | **97.4±2.3** | **99.1±0.8** | 93.5±3.7 | **95.9±2.7** |
| *Improve (over Point-MAE)* | +1.1 | +1.3 | +0.9 | +0.9 |
| *with Cross-Modal Self-Supervised Representation Learning* | | | | |
| ACT [9] | 96.8±2.3 | 98.0±1.4 | 93.3±4.0 | 95.6±2.8 |
| Joint-MAE [14] | 96.7±2.2 | 97.9±1.9 | 92.6±3.7 | 95.1±2.6 |
| I2P-MAE [51] | 97.0±1.8 | 98.3±1.3 | 92.6±5.0 | 95.5±3.0 |
| TAP [38] | 97.3±1.8 | 97.8±1.9 | 93.1±2.6 | 95.8±1.0 |
| ReCon [27] | 97.3±1.9 | 98.9±1.2 | 93.3±3.9 | 95.8±3.0 |

**Object part segmentation.** We conduct part segmentation experiments on the ShapeNetPart [46] to validate the effectiveness of our PCP-MAE method. ShapeNetPart is used to evaluate the learning capacity of models toward knowledge of detailed shape semantics within 3D objects. It comprises 16,881 objects across 16 categories. In alignment with prior research [24], we sample 2048 points from each input point cloud and partition them into 128-point patches. As indicated in Tab. 4, our PCP-MAE yields comparable performance over peer methods and outperforms Point-MAE by a large margin. Particularly, PCP-MAE achieves 84.9% in Cls.mIoU and improves Point-MAE by 0.7%.

**3-D scene segmentation.** Scene segmentation is a challenging task, especially in large-scale 3-D scenes, as it demonstrates the ability of models to comprehend contextual semantics and intricate local geometric relationships. We conduct 3-D scene segmentation on the S3DIS dataset [2], which provides densely annotated semantic labels for point clouds. The results can be observed in Tab. 4. PCP-MAE shows better ability under scene segmentation scenarios and outperforms Point-MAE by 1.1% and 0.5% in mAcc and mIoU, respectively.

### 4.3 Ablation studies

We conducted several experiments to demonstrate the efficacy of PCP-MAE, with results on three variants of ScanObjectNN (%) reported, maintaining experimental settings consistent with Sec. 4.2. PCP-MAE is pre-trained on ShapeNet before fine-tuning on ScanObjectNN.

**Main components in the PCP-MAE.** Compared to Point-MAE which only performs point cloud reconstruction, our PCP-MAE also adopts reconstruction as a pre-training target and differs from Point-MAE mainly in two aspects: 1) Adding PCM which shares parameters with the encoder to guide the encoder to learning to predict centers for masked patches. 2) Instead of directly providing the ground truth $\mathbf{PE}_m$, we replace it with the predicted results $\mathbf{PE}_m^{pred}$. To validate the effectiveness of these two operations in PCP-MAE, we conduct experiments by varying the utilization of PCM and the predicted $\mathbf{PE}_m^{pred}$ on the performance of our PCP-MAE. The Tab. 5 presents the results, showing that only adopting predicting centers as pertaining target outperforms the from-scratch baseline which means the task using PCM proposed by us is meaningful as an isolated pre-training task. Results also indicate that while guiding encoder learning to predict centers improves the performance (w/ PCM), replacing $\mathbf{PE}_m$ with $\mathbf{PE}_m^{pred}$ further enhances the performance.

**Predicting Targets.** The centers can be expressed in two forms: coordinates with a 3-dimensional representation (x, y, z) and positional embedding (PE) with a $D$-dimensional representation obtained by embedding the coordinates, where $D$ refers to the dimensionality of the model. Therefore, there are at least two available targets for center prediction. We claim that the PE form is more suitable

Table 4: Segmentation Results on ShapeNetPart and S3DIS Area 5: Mean intersection over union for all classes Cls.mIoU (%) and all instances Inst.mIoU (%) for Part Segmentation; Mean accuracy mAcc (%) and IoU mIoU (%) for Semantic Segmentation.

| Methods | Part Seg. | | Semantic Seg. | |
|---|---|---|---|---|
| | Cls.mIoU | Inst.mIoU | mAcc | mIoU |
| PointNet [25] | 80.4 | 83.7 | 49.0 | 41.1 |
| PointNet++ [26] | 81.9 | 85.1 | 67.1 | 53.5 |
| DGCNN [37] | 82.3 | 85.2 | - | - |
| PointMLP [23] | 84.6 | 86.1 | - | - |
| *with Single-Modal Self-Supervised Representation Learning* | | | | |
| Transformer [34] | 83.4 | 84.7 | 68.6 | 60.0 |
| CrossPoint [1] | - | 85.5 | - | - |
| Point-BERT [47] | 84.1 | 85.6 | - | - |
| MaskPoint [20] | 84.4 | 86.0 | - | - |
| Point-MAE [24] | 84.2 | 86.1 | 69.9 | 60.8 |
| PointGPT [4] | 84.1 | 86.2 | - | - |
| Point-FEMAE [48] | **84.9** | **86.3** | - | - |
| **PCP-MAE** | **84.9** | 86.1 | 71.0 | **61.3** |
| *Improve (over Point-MAE)* | +0.7 | +0.0 | +1.1 | +0.5 |
| *with Cross-Modal Self-Supervised Representation Learning* | | | | |
| ACT [9] | 84.7 | 86.1 | **71.1** | 61.2 |
| ReCon [27] | 84.8 | 86.4 | - | - |

Table 5: Effects of the main components in the proposed PCP-MAE include the pre-training targets (point cloud reconstruction, learning to predict centers with PCM) and the operation that replaces the $\mathbf{PE}_m$ with predicted centers $\mathbf{PE}_m^{pred}$. The default setting is marked in blue.

| reconstruction | w/ PCM | using $\mathbf{PE}_m^{pred}$ | OBJ_BG | OBJ_ONLY | PB_T50_RS |
|---|---|---|---|---|---|
| ✗ | ✗ | ✗ | 90.01 (scratch) | 88.64 (scratch) | 83.93 (scratch) |
| ✗ | ✓ | ✗ | 92.42 | 92.42 | 88.13 |
| ✗ | ✓ | ✓ | 91.73 | 92.42 | 88.37 |
| ✓ | ✗ | ✗ | 92.94 | 92.42 | 88.65 |
| ✓ | ✗ | ✓ | 90.01 | 88.64 | 84.73 |
| ✓ | ✓ | ✗ | 94.32 | 93.11 | 89.38 |
| ✓ | ✓ | ✓ | 95.52 | 94.32 | 90.35 |

Table 6: Effects of predicting targets.

| predicting targets | OBJ_BG | OBJ_ONLY | PB_T50_RS |
|---|---|---|---|
| coordinates | 94.32 | 92.25 | 88.68 |
| sin-cos PE | 95.52 | 94.32 | 90.35 |

Table 7: Effects of stop-gradient operation.

| stop-gradient | OBJ_BG | OBJ_ONLY | PB_T50_RS |
|---|---|---|---|
| ✗ | 92.42 | 91.56 | 87.36 |
| ✓ | 95.52 | 94.32 | 90.35 |

in this situation because it provides sparser positional information compared to the 3-dimensional coordinates, and it represents a high-dimensional semantic space that is easier and clearer for the model to learn. The experiment results reported in Tab. 6 substantiate this claim.

**Stop-gradient operation.** To prevent the decoder from finding shortcuts during pre-training, we apply a stop-gradient operation $\text{sg}(\cdot)$ to the predicted centers before feeding them into the decoder. This stops the back-propagation of gradients to the PCM, ensuring that any reduction in reconstruction loss does not falsely adjust the predicted centers $\mathbf{PE}_m^{pred}$, as the reconstruction loss only contains information for points in patches but no semantic information for the centers. Experimental results, shown in Tab. 7, indicate a significant performance drop when the stop-gradient operation is omitted.

## 5 Conclusion

In this paper, we first identify the disparities in positional embeddings between 2-D vision and 3-D point cloud, finding that direct leakage of masked centers makes the pre-training in Point-MAE and existing MAE-based methods trivial as shown in Fig. 1. To address this issue, we introduce PCP-MAE, which adds a novel objective to guide the encoder to predict positional embeddings for

the masked centers, enabling the encoder to learn much more semantic representations. A Predicting Center Module which uses cross-attention and shares parameters with the encoder is proposed to achieve this. Exhaustive experiments on SSL benchmarks showcase the superior performance of our method compared to Point-MAE, setting new SOTA results across various tasks.

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

Table 8: Training details for pretraining and downstream fine-tuning.

| Config | ShapeNet | ScanObjectNN | ModelNet | ShapeNetPart | S3DIS |
|---|---|---|---|---|---|
| optimizer | AdamW | AdamW | AdamW | AdamW | AdamW |
| learning rate | 5e-4 | 2e-5 | 1e-5 | 2e-4 | 2e-4 |
| weight decay | 5e-2 | 5e-2 | 5e-2 | 5e-2 | 5e-2 |
| learning rate scheduler | cosine | cosine | cosine | cosine | cosine |
| training epochs | 300 | 300 | 300 | 300 | 60 |
| warmup epochs | 10 | 10 | 10 | 10 | 10 |
| batch size | 128 | 32 | 32 | 16 | 32 |
| drop path rate | 0.1 | 0.2 | 0.2 | 0.1 | 0.1 |
| number of points | 1024 | 2048 | 1024 | 2048 | 2048 |
| number of point patches | 64 | 128 | 64 | 128 | 128 |
| point patch size | 32 | 32 | 32 | 32 | 32 |
| augmentation | Scale&Trans+Rotation | Rotation | Scale&Trans | - | - |
| GPU device | V100 / RTX 3090 | V100 / RTX 3090 | V100 / RTX 3090 | RTX 3090 | RTX 3090 |

Table 9: Different Loss functions. The accuracy (%) on three variants of ScanObjectNN is reported. The default setting is marked in  blue .

| Loss function | OBJ_BG | OBJ_ONLY | PB_T50_RS |
|---|---|---|---|
| $l_1$ Distance | 94.49 | 93.28 | 89.76 |
| Smooth $l_1$ | 95.18 | 92.94 | 89.48 |
| Cosine Similarity | 94.83 | 93.45 | 89.52 |
| $l_2$ Distance | 95.52 | 94.32 | 90.35 |

# Appendix

## A  Additional Experimental Eetails

**Pre-training details.** We use ShapeNet [3] as the pretraining dataset. ShapeNet is a clean set of 3D CAD object models with rich annotations, including 51K unique 3D models from 55 common object categories. The overall pretraining includes 300 epochs, and we use a cosine learning rate [21] of 5e-4 warming up for 10 epochs. AdamW optimizer [22] is used, and the batch size is 128. We run all experiments with single GPU either using RTX 3090 (24GB) or V100 (32GB). More details are shown in Tab. 8.

**Model details.** The model is shown in the Fig. 2. The encoder comprises 12 Transformer blocks with 6 attention heads, while the decoder consists of 4 Transformer blocks with 6 attention heads each. The projector is composed of MLP layers, LayerNorm, ReLU, and another MLP layer, sequentially. The PCM (Predicting Center Module) also comprises 12 Transformer blocks and shares parameters with the encoder by leveraging the same layers within the Transformer block. They differ in their attention mechanisms, *i.e.*, employing self-attention in the encoder or cross-attention in the PCM and their attention objectives.

## B  Additional Ablation Study

**Masking ratio.** We conduct experiments using various masking ratios to determine the most suitable option for our PCP-MAE. The results are illustrated in Fig. 3, and we identify a masking ratio of $m = 0.6$ as optimal.

**Loss function for predicting centers.** The pre-training objective guiding the encoder to predict centers entails a regression task. We explore various regression loss functions to select a suitable loss function for our PCP-MAE. The results are reported in Tab. 9. Notably, the $L_2$ Distance loss emerges as the best choice. Consequently, we adopt the $L_2$ Distance loss for center prediction.

**Pre-training augmentation.** The pre-training augmentation plays an important role in increasing the variance of training samples and thus affecting the performance of our PCP-MAE. We experiment with different augmentations for pre-training and validate their performance through fine-tuning.

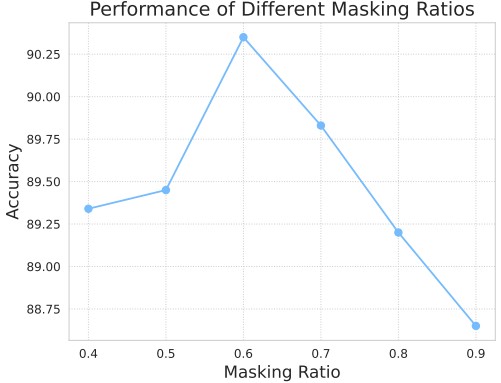
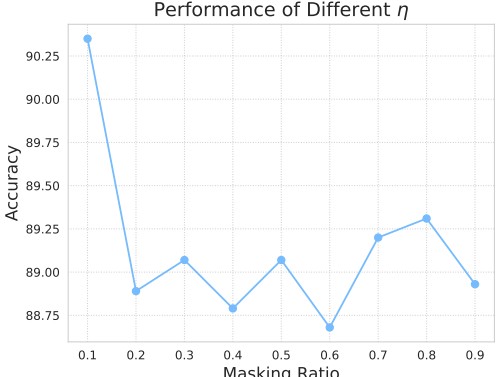

Figure 3: Performance of different masking ratios in our PCP-MAE. The accuracy (%) on PB_T50_RS variant of ScanObjectNN are reported. Masking ratio $0.6$ performs the best.

Figure 4: Performance of different $\eta$ in objective function $\mathcal{L} = \eta\mathcal{L}_{PC} + \mathcal{L}_{Recon}$. The accuracy (%) on PB_T50_RS variant of ScanObjectNN are reported. $\eta = 0.1$ performs the best.

Table 10: Effects of pre-training augmentations. The accuracy (%) on three variants of ScanObjectNN is reported. The default setting is marked in blue.

| Augmentation | OBJ_BG | OBJ_ONLY | PB_T50_RS |
|---|---|---|---|
| - | 94.19 | 92.08 | 88.06 |
| Jitter | 93.63 | 92.25 | 88.05 |
| Horizontal Flip | 93.45 | 91.91 | 88.82 |
| Scale&Translate | 94.14 | 92.25 | 88.54 |
| Rotation | 94.14 | 93.11 | 88.96 |
| Rotation + Scale&Translate | 94.83 | 93.11 | 90.18 |
| Scale&Translate + Rotation | 95.52 | 94.32 | 90.35 |

When fine-tuning, we adopt Rotation for augmenting ScanObjectNN. The results, presented in Tab. 10, show that the combination of augmentations, particularly Scale & Translate + Rotation, increases the diversity of training samples and yields the best performance. We adopt it as the augmentation for the pre-training of our PCP-MAE.

**Additional analysis on the data augmentation.** We used the Scale&Translate+Rotation augmentation for pre-training because this combination benefits our PCP-MAE, differing from previous baselines. Since we apply different pre-training augmentations compared to prior methods, we report the performance of the baseline (Point-MAE [24]) under various augmentations in Tab. 11 for clarity. Additionally, we test previous SOTA methods (Point-FEMAE [48] and ReCon [27]) with our augmentation strategy and observe a slight performance drop, as shown in Tab. 12. These results indicate that the explored augmentation composition benefits our methods more than others. **This improvement can be attributed to the fact that the combination of augmentations enables a variety of centers, allowing our PCP-MAE to learn a richer distribution of center information and become more robust.** In contrast, the baseline method Point-MAE benefits only slightly from this augmentation due to relying solely on point cloud reconstruction during pre-training, and prior SOTA methods like ReCon and Point-FEMAE do not empirically benefit from it.

**Different $\eta$ in the objective function.** As discussed in Sec. 3.4, the objective function includes a scaling factor $\eta$ to balance the point reconstruction loss $\mathcal{L}_{Recon}$ and the center prediction loss $\mathcal{L}_{PC}$, formulated as $\mathcal{L} = \eta\mathcal{L}_{PC} + \mathcal{L}_{Recon}$. As depicted in Fig. 4, the optimal performance is achieved when $\eta = 0.1$. Therefore, we adopt this value as the scaling factor for the objective function.

**Projector for predicting centers.** After obtaining the output of the last transformer block in PCM, it is passed through a projector comprising MLP layers, ReLU, and LayerNorm to predict masked centers. In contrast, the output of the encoder is directed into a decoder, which consists of transformer blocks, for further reconstructing the point cloud. Thus, we conduct experiments to investigate whether additional transformer layers are necessary for PCM to enhance the decoding of the predicted

Table 11: Performance comparison of Point-MAE with different augmentations. The star (*) marks the setting used by the original Point-MAE [24], the dagger ($\dagger$) by peer SOTA methods such as Point-FEMAE [48] and ReCon [27], and the double dagger ($\ddagger$) by us.

| Augmentation | | ScanObjectNN | | |
|---|---|---|---|---|
| Pre-training | Fine-tuning | OBJ-BG | OBJ-ONLY | PB-T50-RS |
| Scale&Translate | Scale&Translate * | 90.02 | 88.29 | 85.18 |
| Rotation | Rotation $\dagger$ | 92.60 | 91.91 | 88.42 |
| Scale&Translate+Rotation | Rotation $\ddagger$ | 92.94 | 92.25 | 88.86 |

Table 12: The performance of previous SOTA methods using our explored augmentation.

| Method | OBJ-BG | OBJ-ONLY | PB-T50-RS |
|---|---|---|---|
| Point-FEMAE (Origin) | 95.18 | 93.29 | 90.22 |
| Point-FEMAE (Our augmentation) | 94.32 | 92.94 | 89.38 |
| ReCon (Origin) | 95.18 | 93.63 | 90.63 |
| ReCon (Our augmentation) | 94.49 | 92.77 | 89.55 |

Table 13: Number of transformer layers in the projector. The accuracy (%) on three variants of ScanObjectNN is reported. The default setting is marked in blue .

| Depth | # P (M) | OBJ_BG | OBJ_ONLY | PB_T50_RS |
|---|---|---|---|---|
| 0 | 29.05 | 95.52 | 94.32 | 90.35 |
| 1 | 31.27 | 94.14 | 92.94 | 88.93 |
| 2 | 33.05 | 94.49 | 93.45 | 90.04 |
| 3 | 34.81 | 93.28 | 92.25 | 89.27 |
| 4 | 36.59 | 94.49 | 93.80 | 89.55 |

representations. The results presented in Tab. 13 indicate that additional transformer layers do not improve performance, hence we continue to employ a simple projector for decoding the predicted representations obtained by the PCM.

**Shared parameters between the encoder and the PCM.** We conduct an ablation study to compare the effects of sharing parameters versus not sharing parameters between the encoder and the PCM. As shown in Tab. 14, sharing parameters not only significantly reduces the number of parameters but also enhances the performance of the model.

This improvement can be attributed to the following reasons: The encoder is responsible for encoding visible tokens to obtain semantic representations and is the only component used during fine-tuning. When the encoder and PCM share parameters, updates from the PCM are directly incorporated into the encoder. This allows the encoder (or PCM) to learn more semantic representations and more effectively understand the relationship between visible representations and masked patches. Although the loss for predicting centers back-propagates to the encoder due to the cross-attention between the PCM and encoder, its impact on the encoder is indirect when parameters are not shared. In this case, the parameters of the PCM are useless and wasted in downstream tasks because only the encoder is utilized during fine-tuning. In conclusion, sharing parameters makes the updates of the PCM directly influence the encoder, leading to more efficient and effective learning and enabling the encoder to learn more semantic representations.

Concerning the above advantages, we choose to share weights between the encoder and the PCM in our method.

## C   Additional Visualization Results

After pre-training Point-MAE with a 100% mask ratio, *i.e.*,, the encoder is completely discarded during training and only the decoder is utilized, the decoder is able to reconstruct the full point

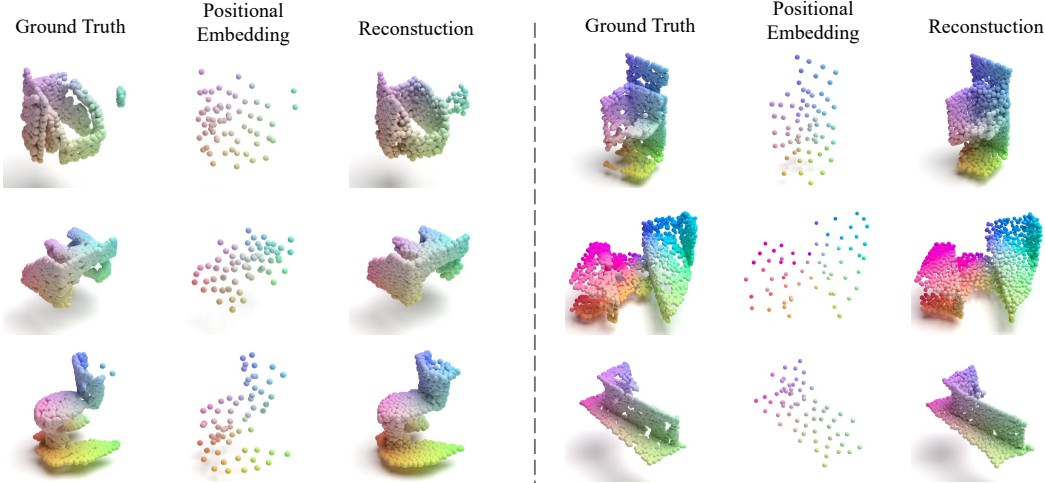

Figure 5: Additional visualization results of Point-MAE reconstruction results on the ScanObjectNN dataset.

Table 14: Ablation on shared parameters between the encoder and PCM. The accuracy (%) on three variants of ScanObjectNN is reported. The default setting is marked in blue.

| Type | # P (M) | OBJ_BG | OBJ_ONLY | PB_T50_RS |
|------|---------|--------|----------|-----------|
| shared | 29.05 | 95.52 | 94.32 | 90.35 |
| non-shared | 50.77 | 94.66 | 92.77 | 89.27 |

cloud using only masked positional embeddings, as shown in Fig. 1. We refer to this as the "masked center leakage", which demonstrates that the decoder does not necessarily rely on the encoder's representation during the pre-training of Point-MAE.

To extend the validation of this phenomenon beyond the ShapeNet dataset [3], which primarily contains objects with clear and well-defined shapes, we conducted experiments on another dataset, ScanObjectNN [33]. This dataset poses a more significant challenge, as it consists of real-world scans that include background clutter and occlusions. The visualization results in Fig. 5 show that the masked center leakage phenomenon is also present in this more complex dataset, further demonstrating the generality of the phenomenon that we identified.

## D  Limitations and Future Works

PCP-MAE is a simple and effective method that improves Point-MAE by additionally learning to predict the centers, which significantly boosts performance. However, there are some limitations in PCP-MAE, which may be two-fold. PCP-MAE is a single-modal self-supervised method. Nevertheless, the current dataset of 3-D point clouds is constrained in size due to the challenges associated with collecting point cloud data, which in turn limits the wider applicability of our approach. Additionally, while PCP-MAE capitalizes on generative learning, it does not leverage the benefits of contrastive learning and some other works such [27, 7] show great performance by harmoniously combining generative learning [56, 53] and contrastive learning [54, 55, 52].

To address these issues, future works could focus on developing a multi-modal PCP-MAE, explainable AI [7, 6, 5, 8] or a hybrid model that effectively combines the strengths of both generative and contrastive learning.

## E  Broader Impacts

Our PCP-MAE demonstrates a marked improvement over the existing Point-MAE and establishes new state-of-the-art (SOTA) benchmarks across a variety of tasks in point cloud understanding. This

enhancement is pivotal in advancing technologies reliant on precise spatial recognition [43, 42, 44, 45, 6], such as autonomous driving, which demands accurate environmental perception for safety and efficiency. Moreover, the versatility of the PCP-MAE extends its utility to other critical applications, including urban planning, where detailed and scalable 3D city modeling is essential, and in augmented reality, enhancing interactive experiences by integrating more accurate virtual information with the real world. While our PCP-MAE marks an advancement in point cloud understanding with broad applications ranging from autonomous driving to urban planning, it is also susceptible to potential negative societal impacts. The primary concerns stem from its potential use in surveillance systems, where enhanced spatial recognition capabilities could lead to privacy infringements.

