# OpenReview forum: "PCP-MAE: Learning to Predict Centers for Point Masked Autoencoders"
_NeurIPS.cc/2024/Conference — NeurIPS 2024 spotlight_

### Official Review · Reviewer_rD3v · 2024-07-10

**Soundness:** 3
**Presentation:** 3
**Contribution:** 4
**Rating:** 6
**Confidence:** 4

**Summary:**

This paper makes a novel observation (at least to me) that for mask-based autoencoder paradigm for point cloud self-supervised pretraining, the centers of patches are important and the reconstruction objective does not necessarily rely on representations of the encoder. This is different from the 2-D case on mask-based autoencoding for images. Then the authors introduce a simple yet strong scheme that directly learning to Predict Centers for Point Masked AutoEncoders to prevent the encoder from the failure of semantic information learning. This approach is more efficient than baselines, and achieves state-of-the-art performance on public datasets.

**Strengths:**

1) The observation is interesting and the motivaiton behind the approach is clear, though the abstract I think needs improvement to make it more logical, crisp, and convincing.
2) It shows the difference between 2-D and 3-D data for mask-based autoencoding, and the knowledge and evidence shown in this paper is worth publishing (in NeurIPS 2024) at this moment.
3) The devised approach is cost-effective, and shows strong empirical performance.
4) The paper is basically clearly written with comprehensive experiments.

I think the paper gives a very important observation that center-aware objective makes the pre-training trivial. It opens up space for improvement for the community as also reflected by their experimental results. And this is the major reason for me to vote for acceptance.

**Weaknesses:**

The authors may discuss the limitation of their approach (if there was any).

The abstract can be improved, e.g. the numbers in abstract is vague as there is no specific metric.

**Questions:**

Could this approach inspire the development in other areas of selfsupervised learning?

---

> ### Author Rebuttal · Authors · 2024-08-03
>
> Thank you for your time, detailed comments, and valuable suggestions. We are delighted that you recognize the clear motivation, high efficiency, and novelty of our PCP-MAE. Here are our responses to the questions you raised:
>
> > Q1. The limitation of our approach.
>
> We have discussed the limitations of our approach in Appendix C of the submitted paper. The limitations of our work can be summarized as follows:
>
> 1. PCP-MAE is a single-modal self-supervised method. However, the current dataset of 3D point clouds is constrained in size due to the challenges associated with collecting point cloud data, which in turn limits the wider applicability of our approach.
> 2. PCP-MAE capitalizes on generative learning but does not leverage the benefits of contrastive learning. Some other works, such as ReCon [1], show great performance by harmoniously combining generative and contrastive learning.
>
> Future work could focus on developing a multi-modal PCP-MAE or a hybrid model that effectively combines the strengths of both generative and contrastive learning.
>
> > Q2. The abstract can be improved, e.g. the numbers in abstract is vague as there is no specific metric.
>
> Thank you for your valuable advice. We have enhanced the abstract by adding a specific metric.
>
> Regarding lines 17-20, we have modified the text from
> - "Our method is of high pre-training efficiency compared to other alternatives and achieves great improvement over Point-MAE, particularly outperforming it by 5.50%, 6.03%, and 5.17% on three variants of ScanObjectNN."
>
> to
>
> - "Our method is of high pre-training efficiency compared to other alternatives and achieves great improvement over Point-MAE, particularly surpassing it by 5.50% on OBJ-BG, 6.03% on OBJ-ONLY, and 5.17% on PB-T50-RS for 3D object classification on the ScanObjectNN dataset."
>
> If you have any further concerns, feel free to contact us.
>
> > Q3. Could this approach inspire the development in other areas of self-supervised learning?
>
> Yes. Our approach offers insights for the broader field of self-supervised learning. Specifically, it challenges the conventional use of positional encodings in the reconstruction process, highlighting the importance of semantic richness in these encodings in the point cloud SSL. While our study focuses on point cloud data in 3D form, the implications extend to other 3D data representations such as meshes and voxels. In these contexts, the usage of positional encodings may need to be reconsidered if they contains case-specific information rather than only case-agnostic information.
>
> Furthermore, our findings illuminate considerations for adapting methods across different domains. For example, when extending the Masked Autoencoder (MAE) approach to 3D from 2D, it is crucial to account for the distinct interpretations of positional encodings in these dimensions. This underscores the need for domain-specific adjustments to ensure the transferability and effectiveness of self-supervised learning techniques.
>
>
> [1] Qi Z, Dong R, Fan G, et al. Contrast with reconstruct: Contrastive 3d representation learning guided by generative pretraining[C], ICML 2023.

---

### Official Review · Reviewer_GmbZ · 2024-07-13

**Soundness:** 3
**Presentation:** 3
**Contribution:** 2
**Rating:** 4
**Confidence:** 5

**Summary:**

The paper proposes a novel self-supervised learning method called PCP-MAE for point cloud understanding. The key innovation of PCP-MAE is that it guides the model to learn to predict the positional embeddings of the centers of masked regions, rather than directly providing the coordinates. This approach encourages the encoder to learn richer semantic representations, leading to performance improvements on downstream tasks such as point cloud classification.

**Strengths:**

1.Instead of directly providing the coordinates of the centers of masked regions, PCP-MAE guides the model to learn to predict the positional embeddings of these centers. This idea encourages the encoder to learn richer semantic representations.

2.The authors have conducted a comprehensive set of experiments, comparing PCP-MAE against state-of-the-art methods on a variety of point cloud understanding tasks.

**Weaknesses:**

1.There are significant concerns about the actual effectiveness of the center point prediction method being discussed. Although the author conducted numerous experiments to demonstrate the superiority of their method, these comparisons with previous methods are unfair. This unfairness is reflected in the following aspects: 1. Although this method has achieved significant improvement, it seems that the main improvement comes from the use of more complex data augmentation tricks during pre training, without a fair comparison with previous methods when using the same tricks. Due to the lack of detailed code provided by the author, based on my personal experience, when using only the center point prediction method with the same settings or tricks, the proposed method does not show significant improvement and may even experience performance degradation. For example, when using the center point of ground truth without center point prediction and with the same settings such as position encoding and data augmentation, I don't think the author's center point prediction will be better than using the ground truth method.

2.On line 69, the author get the conclusion that 'the reconstructing objective may make the encoder unable to learn semantic features'. I would like to know how the author came to this conclusion. Can a randomly initialized token with a 100% mask rate be used to reconstruct the approximate shape of a point cloud using only the center point as the position encoding through an encoder? I don't think so. This phenomenon does not mean that the encoder cannot learn semantic features. Secondly, the author only conducted a qualitative analysis without providing a detailed quantitative comparison. In fact, the CD distance of the point cloud reconstructed by this method is much greater than that of the Point MAE method, indicating that the effect of the point cloud reconstructed by this method is very poor.

3.The author's improvement is incremental and limited, so I do not think that this paper meets the contribution criteria of NeurIPS. The proposed method is only an improvement on Point MAE, in fact, Point MAE is just one method in the field of point cloud self supervised learning.

4.The author did not demonstrate the generality of the proposed method. Assuming that this central prediction method is effective, it should be applicable to all MAE based point cloud self-supervised methods, such as Point BERT, ACT, Point-M2AE, PointFEMAE, I2P-MAE, Recon, etc. The author lacks further analysis of the generality of the proposed ideas.

5.Insufficient Visualizations and Explanations: The paper lacks some key visualizations and explanations that would aid reader understanding. For example, the pipeline (Fig. 2) does not clearly annotate the "stop gradient" operation, which is a crucial component of the proposed method. Additional visualizations and explanations would help readers better comprehend the technical details of PCP-MAE.

**Questions:**

Although the author has demonstrated superior improvement in various experiments, this improvement seems to come more from trick's improvements (such as data augmentation) rather than the proposed method itself. Meanwhile, the author lacks a detailed analysis of the generality of the proposed method. Therefore, I reject this article and encourage the author to conduct a more in-depth, thorough, and fair analysis and comparison.

**Limitations:**

The author has pointed out some issues in the Limitations section, and I hope the author can make more improvements based on the review comments, so that their work becomes truly solid rather than superficial incremental improvements.

---

> ### Author Rebuttal · Authors · 2024-08-07
>
> Thank you for taking the time to review our paper. There may be some misunderstandings on your part regarding certain aspects of our work. We hope that our response will address your concerns. Here is our clarification.
> > Q1.1. Comparisons with previous methods are unfair. It seems that the main improvement comes from the use of more complex data augmentation tricks during pre training, without a fair comparison with previous methods when using the same tricks.
>
> Thank you for pointing out the lack of baseline performance using the same augmentation. Firstly, different methods suit different pre-training augmentations. During experimentation, we found our approach works well when using a composition of augmentations, *i.e.*, Scale&Translate+Rotation. This can be attributed to the fact that a composition of augmentations enables a variety of centers, thus allowing our PCP-MAE to learn richer center distribution information and be more robust. While this also slightly benefits the performance of the baseline Point-MAE, **it doesn't benefit the previous SOTA methods such as Point-FEMAE and ReCon.** The performance of Point-MAE under different augmentations is displayed as follows for clarity:
> Augmentation||OBJ-BG|OBJ-ONLY|PB-T50-RS
> -|-|-|-|-
> Pre-training|Fine-tuning|
> Scale&Translate|Scale&Translate $^*$ |90.02|88.29|85.18
> Rotation|Rotation $^\dagger$ |92.60|91.91|88.42|
> Scale&Translate+Rotation|Rotation $^\ddagger$|92.94|92.25|88.86
>
> $^*$ Adopted by Point-MAE.
>
> $^\dagger$ Adopted by other SOTA methods include Point-FEMAE and ReCon.
>
> $^\ddagger$ Adopted by our PCP-MAE.
>
> **When aligning the augmentations of Point-FEMAE and ReCon with ours, there are no performance gains, and they even suffer from a performance drop.** This phenomenon is caused by the features of different methods. The performance is shown as follows:
> Method|OBJ-BG|OBJ-ONLY|PB-T50-RS
> -|-|-|-
> Point-FEMAE (Origin)|95.18|93.29|90.22
> Point-FEMAE (Our augmentation)|94.32|92.94|89.38
> ReCon (Origin)|95.18|93.63|90.63
> ReCon (Our augmentation)|94.49|92.77|89.55
>
> **Feel free to validate this through their official codebase.** We apologize for the lack of baseline performance in our paper, which we will definitely add in our next version to ensure fairness and clarity. However, it's also important to reclaim that the composition of augmentations benefits our method due to its features and hampers other SOTA methods, which means the main improvement comes from our approach, not only from the explored augmentations.
> > Q1.2. Through your personal experience, you found when using only the center point prediction method with the same settings or tricks, the proposed method does not show significant improvement and may even experience performance degradation.
>
> First, when you try our idea in person, please note there remain some subtle designs and pay attention not to implement them incorrectly. These include sin-cos positional embedding followed by MLP projection, the detailed design of the projector, the shared weight encoder with self/cross attention and *etc*.
>
> Second, regarding the $\eta$ in line 210, it's important to select an appropriate $\eta$ to improve the performance. Empirically, a small $\eta$ is preferred, which still needs to be tuned carefully. So be careful to choose an appropriate $\eta$ when you try our idea.
>
> Moreover, the sin-cos position embedding doesn't bring performance gain:
> Method| OBJ-BG | OBJ-ONLY | PB-T50-RS
> -|-|-|-
> Point-MAE (Orig.)|92.94|92.25|88.86
> Point-MAE (Sin-cos)|92.94| 92.42 | 88.65
>
> The pre-training augmentation aligns well with our approach and may hinder other methods. This should be seen not as a trick but as a feature of our PCP-MAE. For more details, please refer to our response to Q1.1
>
> **Regarding the effectiveness of our approach, the ablation results have confirmed its effectiveness when all other experimental setups are aligned:**
> Reconstruction|w/PCM|Using $PE_m^{\text{pred}}$ |OBJ_BG|OBJ_ONLY|PB_T50_RS
> -|-|-|-|-|-
> ✓|✗|✗|92.94|92.42|88.65
> ✓|✓|✗|94.32|93.11|89.38
> ✓|✓|✓|**95.52**|**94.32**|**90.35**
>
> **We have provided our code and the corresponding checkpoints via an anonymous link to the AC.** Hope this will help you to better comprehend our method.
>
> > Q2.1. Can a randomly initialized token with a 100% mask rate be used to reconstruct the approximate shape of a point cloud using only the center point as the position encoding through an encoder?
>
> Yes. When only given positional embeddings for a masked patch and a global, randomly initialized **learnable** mask token, the point cloud can be reconstructed only using the decoder after pre-training. The visualization can be seen in Fig. 1. Furthermore, we found that this phenomenon also holds on a more challenging dataset, ScanObjectNN.
>
> > Q2.2. This phenomenon does not mean that the encoder cannot learn semantic features.
>
> This phenomenon indicates that the decoder does not necessarily rely on the representations of the encoder for reconstruction, which **may** make the encoder unable to learn semantic representations. Our PCP-MAE guides the encoder to learn to predict semantically rich centers to force the learning of richer representations, thus improving performance.
>
> > Q2.3. The quantitative reconstruction analysis between Point-MAE and Mask 100%.
>
> We use the Reconstruction Chamfer Distance L2 loss to evaluate the quality of reconstruction. The losses for them are shown as follows:
> Methods|Epoch 1 Loss|Epoch Last Loss
> -|-|-|
> Point-MAE|0.12921|0.00270
> Mask 100|0.15823|0.00324
>
> The Mask 100% exhibits a slightly higher loss than that of the original Point-MAE, but continues to decline and can reconstruct the point cloud after pre-training. Though the encoder slightly benefits point cloud reconstruction, the reconstruction can still be achieved without it. This phenomenon aligns with our core idea that the decoder does not necessarily rely on the representations of the encoder.
>
> **Please check the official comment for the remaining questions.**

---

> > ### Comment · Reviewer_GmbZ · 2024-08-07
> > **Quick Response to author**
> >
> > I believe you could provide me with your anonymous checkpoints and source code. I hope to verify the effectiveness of your method through a detailed analysis of your code.

---

> ### Author Response · Authors · 2024-08-07
> **Continued rebuttal**
>
> > Q3.1. The author's improvement is incremental and limited.
>
> We have demostrated the effectiveness and superior performance of our method through ablations and experiemnts. Hope the rebuttal can help you better understand our method.
>
> > Q3.2. The proposed method is only an improvement on Point-MAE, in fact, Point MAE is just one method in the field of point cloud self supervised learning.
>
> We agree with you that Point-MAE is just one method in 3D SSL field, but it's of significant meaning in this field. It inspires many great works, improving it from different aspects including Point-M2AE, Point-FEMAE, ACT, Joint-MAE, ReCon and etc. Our method is proposed based on a motivating observation to improve the milestone method.
>
> We acknowledge that Point-MAE represents just one of many approaches within the field of 3D self-supervised learning. However, its contribution cannot be overstated. Point-MAE has served as a foundational model, sparking a wave of innovative research and developments aimed at enhancing its framework. This includes advancements through models like Point-M2AE, Point-FEMAE, ACT, Joint-MAE, and ReCon, among others. Our proposed method builds upon these insights, originating from a critical observation that seeks to further refine this pivotal technique.
>
> > Q4. The generality of the proposed method. And our approach should be applicable to all MAE based point cloud self-supervised methods.
>
> Thank you for proposing a good question to inspire us to explore the generality of our approach. You mensioned that our method should fits all MAE-based method. However, we design our method to improve Point-MAE but not implement an universal framework. Take Point-FEMAE as an example. It address the limited representation issue in the Point-MAE but that doesn't mean it fits all MAE-based method and can improve their performance undoubtedly.
>
> Intuitively, our method fits all MAE-based method, but the concrete design should be carefully considered. Because different MAE-based method may optimize towards different directions, though within the same framework. Directly add PCM (Predicting Center Module) may hamper the consistency.
>
> To explore the generality of our approach, **we incorporated PCM into MAE-based methods including Point-BERT and Point-FEMAE with additional minor changes to maintain architecture consistency.** The experiment results show that incorporating our center point predicting task enhances performance when other experiment setups are aligned:
>
> Method|OBJ-BG|OBJ-ONLY|PB-T50-RS
> -|-|-|-|
> Point-BERT|87.43|88.12|83.07
> Point-BERT+PCP ($\eta=2.0$)|89.32|89.84|84.77|
> Point-FEMAE $^\dagger$|94.32|92.94|89.38
> Point-FEMAE+PCP ($\eta=0.1$) $^\dagger$|95.00|93.45|89.83|
>
> $^\dagger$ Reproduced results adopting our augmentation $i.e.$, Scale&Translate+Rotation for pre-training.
>
> Due to time constraints, exploration of various design choices and different $\eta$ was not feasible. We believe that further customization could lead to more substantial improvements.
>
> > Q5. Insufficient Visualizations and Explanations.
>
> Thank you for your valuable advice. We have added the stop gradient to Fig. 2 according to your suggestion. Please check the polished figure in the attached rebuttal PDF. If you have any further concerns, please contact us.

---

> ### Author Response · Authors · 2024-08-07
> **Anonymous source code and checkpoints are available.**
>
> Thank you for your prompt reply and valuable feedback.
>
> As per your suggestion, we have made the comments on the anonymous source code and checkpoints visible to you. You can now access this content at the following link:
> https://openreview.net/forum?id=i1xjK5a0X8&noteId=PtuAJIdLVH
>
> If you have any questions or need further clarification while reviewing, please don't hesitate to contact us at your earliest convenience.

---

> > ### Comment · Reviewer_GmbZ · 2024-08-11
> > **Results reproduction**
> >
> > Dear Authors, Reviewers, and AC,
> >
> > After running the code provided by the authors, I found that the proposed method does indeed show some improvement over the baseline Point-MAE. However, I think the authors have somewhat **overstated their results**. As shown in the following tables, considering that the method does not show improvements over the state-of-the-art and is merely an incremental improvement over Point-MAE with limited generality and minimal inspiration for other research, I am raising my score to a border reject. While I acknowledge that the authors' work has some merit, I consider it to be a gradual contribution and do not believe it meets the NeurIPS acceptance standards.
> >
> > Specifically, I reran the authors' code on the classification task of ScanObjectNN using the recommended settings. I conducted a relatively fair comparison between PCP-MAE, the baseline Point-MAE, and the state-of-the-art method Recon. I repeated the experiments ten times on three different variants of ScanObjectNN, using ten identical random seeds for each method. I reported the average and highest values across the ten runs to further eliminate the impact of randomness. The experiments show that, while PCP-MAE indeed offers some improvement over the original Point-MAE, it underperforms compared to the state-of-the-art method Recon in both average and highest values.
> >
> > Additionally, the authors reported their results in Table 2 as 95.52%, 94.32%, and 90.35%, which differ significantly from the results I reproduced: 94.32%, 93.28%, and 89.94%, respectively. The discrepancies are 1.20%, 1.04%, and 0.41%. Although random seeds can significantly impact the results, a fair comparison using ten identical seeds is sufficient to illustrate the issue. Therefore, I believe the authors have overstated their results.
> >
> > **Best result of ten experiments.**
> >
> > | Methods  |  OBJ-BG |  OBI-ONLY | PB-T50-RS  |
> > | ------------ | ------------ | ------------ | ------------ |
> > | Point-MAE  | 93.12  | 92.77  | 89.04  |
> > | PCP-MAE  | 94.32  | 93.29  | 89.94  |
> > | Recon  |  95.19 | 93.12  | 90.25  |
> >
> > **Average result of ten experiments.**
> >
> > | Methods  |  OBJ-BG |  OBI-ONLY | PB-T50-RS  |
> > | ------------ | ------------ | ------------ | ------------ |
> > | Point-MAE  | 92.67  | 92.31  | 88.59  |
> > | PCP-MAE  | 93.99  | 92.62  | 89.34  |
> > | Recon  |  94.37 | 92.51  | 89.95  |

---

> ### Author Response · Authors · 2024-08-11
> **Response 1**
>
> Dear Reviewer GmbZ,
>
> First, we want to express gratitude for your responsible and thorough review. **We are also pleased to have clarified, to some extent, a previous misunderstanding; specifically, that the improvements observed are attributable to our method, not to a "data augmentation trick."**
>
> Given that there remain some concerns, we would like to address the issues you raised in your comment.
> > Q1. The authors have somewhat overstated their results. The authors reported their results in Table 2 as 95.52%, 94.32%, and 90.35%, which differ significantly from the results I reproduced: 94.32%, 93.28%, and 89.94%, respectively. The discrepancies are 1.20%, 1.04%, and 0.41%.
>
> We agree with you that random seeds can significantly impact experimental results. However, **it is important to consider that variations in hardware, such as different machines and GPU types, as well as differing computational environments, also play a crucial role in the performance outcomes.** This is particularly evident in the field of 3D SSL, where such factors are well-known to affect results. That's to say, discrepancies between the machines used for pre-training and those used for fine-tuning can lead to a performance drop. This is exemplified by our observations that directly downloaded checkpoints from Point-FEMAE and ReCon yield results that are consistently lower than the initially reported outcomes:
> Methods|OBJ-BG|OBJ-ONLY|PB-T50-RS
> -|-|-|-
> ReCon (reported)|95.18|93.63|90.63
> ReCon (reproduced)|95.00|92.94|90.18
> Point-FEMAE (reported)|95.18|93.29|90.22
> Point-FEMAE (reproduced)|94.49|92.77|89.73
>
> To address the discrepancies in your reproduced results and the reported results, we suggest the possibility of machine inconsistency between pre-trained machine (ours) and the fine-tuning machine (yours). For a truly fair comparison, we recommend pre-training our model directly rather than using pre-provided checkpoints to prevent the performance drop brought by the different machines (our machine and your machine).
>
> Additionally, it is quite unexpected that you got such a bad performance with our PCP-MAE. For instance, you reported only a 94.32% accuracy on the OBJ-BG using your setup, whereas we routinely achieve performances exceeding 95% with relative ease. Furthermore, for the OBJ-ONLY, we replicate the methodology Point-MAE used to produce their results, which involves running multiple experiments at checkpoints 300, 275, and 250 (refer to the issues discussed on the official Point-MAE GitHub repository). We achieved 94.15% at ckpt-300 and 94.32% at ckpt-275. Both of these checkpoints easily yield performances above 93.45%, which is higher than the results you obtained.
>
> **We provide the logs for three variants of ScanObjectNN (OBJ-BG, OBJ-ONLY, PB-T50-RS), which we got in April 2024. We have updated these logs at the anonymous code link https://anonymous.4open.science/r/2128-PCP-MAE-529D for you to check. The timestamps in these logs are continuous, indicating that the experiments were conducted continuously and fairly without selectively choosing high-performing logs.** Specifically, there are 10 continuous logs for the OBJ-BG, 6 for the OBJ-ONLY (ckpt-300), 8 for the OBJ-ONLY (ckpt-275), and 100 for the PB-T50-RS. Please review the logs at the provided code link. The specific accuracies from these logs are detailed below:
>
>
> Benchmark|1|2|3|4|5|6|7|8|9|10
> -|-|-|-|-|-|-|-|-|-|-
> OBJ-BG (ckpt-300)|**95.53**|94.14|94.83|94.14|94.32|95.35|94.49|94.83|93.45|94.49
> OBJ-ONLY (ckpt-275)|93.29|92.94|93.29|92.60|93.12|**94.32**|93.63|93.63|||
> OBJ-ONLY (ckpt-300)|93.45|92.77|93.80|93.63|92.59|**94.15**|
> PB-T50-RS (ckpt-300)|89.49|**90.35**|88.86|89.55|89.49|89.83|89.63|89.24|89.83|89.38|
>
> The results of this logs can be summarized as:
>
> Best results:
> Methods|OBJ-BG|OBJ-ONLY||PB-T50-RS
> -|-|-|-|-
> Point-MAE|93.12|92.77||89.04
> PCP-MAE|95.53 (ckpt-300)|94.32 (ckpt-275)|94.15 (ckpt-300)|90.35 (ckpt-300)
> Recon|95.19|93.12||90.25
>
> Average results:
> |Methods|OBJ-BG|OBJ-ONLY||PB-T50-RS|
> |-|-|-|-|-|
> Point-MAE|92.67|92.31||88.59
> PCP-MAE|94.56 (ckpt-300)|93.35 (ckpt-275)|93.40 (ckpt-300)|89.57 (ckpt-300)
> Recon|94.37|92.51||89.95
>
> **The results show that we do not overstate our results.**
>
> We also understand that you directly ran the ReCon pre-training checkpoints on your machine, which may have also introduced architectural inconsistencies, leading to a performance drop, albeit less than ours. Therefore, comparing your results directly with ours might not provide a fair assessment. However, due to the more significant machine and GPU inconsistencies encountered when you use the checkpoints trained on our machine, we recommend that you run the pre-training PCP-MAE yourself and then fine-tune to conduct an entirely fair comparison.
>
> **Finally, we wish to affirm that we are fully responsible for the logs (results) provided and guarantee their reproducibility.** We will definitely release the corresponding fine-tuning checkpoints for each benchmark.

---

> ### Author Response · Authors · 2024-08-11
> **Response 2 (Response 1 continued.)**
>
> > Q2. Considering that the method does not show improvements over the state-of-the-art (ReCon) and is merely an incremental improvement over Point-MAE.
>
> We previously addressed this question in our rebuttal. Initially, our method was motivated by the observation that point cloud reconstruction does not necessarily rely on the encoder's representation. Hence, we introduced the Point Center Masking (PCM) technique to guide the encoder in predicting masked centers effectively.
>
> Our method enhances Point-MAE with reasonable additional computational needs and achieves state-of-the-art (SOTA) performance in some benchmarks. **Although our performance improvements over previous SOTA methods like ReCon are marginal, it is crucial to highlight the distinct advantages of our approach:**
>
> 1. **Independence from Pre-trained Models:** Unlike ReCon, which heavily relies on pre-trained models such as the text encoder from CLIP and ViT-B pre-trained on ImageNet, our PCP-MAE is simpler and more concise, as it does not depend on external pre-trained models. This simplicity facilitates easier implementation in real-world scenarios.
>
> 2. **Single-modal Data Utilization:** The pre-training of ReCon not only requires point cloud data but also paired images and text, whereas our PCP-MAE exclusively utilizes single-modal data, i.e., point cloud. The need for paired data significantly hampers the scalability of the ReCon model, as larger models require larger amounts of data according to the scaling law. To build datasets larger than ShapeNet, it is necessary to collect extensive amounts of unlabeled point cloud data, render paired images, and manually annotate corresponding text descriptors. The rendering process is time-consuming, and annotation is labor-intensive, contradicting the initial intentions of self-supervised learning. Currently, ReCon uses the labels from point cloud samples as text inputs in the ShapeNet pre-training dataset, which somewhat violates the principles of self-supervised learning. **In contrast, our PCP-MAE can scale easily using only unlabeled point cloud data.**
>
> 3. **Enhanced Efficiency:** ReCon introduces greater complexity with more trainable parameters, whereas our PCP-MAE significantly improves efficiency. We've detailed these comparisons in the rebuttal. Here comes the comparison again for clarity:
>
>     Method|Pre-training|| Fine-tuning| | | | |
>     -|-|-|-|-|-|-|-
>     ||Params (M)|Time (s/epoch)|Params (M)|OBJ-BG Time (s/epoch)|OBJ-ONLY Time (s/epoch)|PB-T50-RS Time (s/epoch)|ModelNet40 Time (s/epoch)
>     PCP-MAE (ours)|29.5 (1.00x)|120 (1.00x)|22.1 (1.00x)|10 (1.00x)|10 (1.00x)|49 (1.00x)|30 (1.00x)
>     ReCon|140.9 (4.78x)|452 (3.77x)|43.6 (1.97x)|13 (1.30x)|13 (1.30x)|60 (1.22x)|42 (1.40x)
>
> Though the improvements of our PCP-MAE over the cross-modal method ReCon are not statistically significant, our method notably excels in implementation feasibility and training efficiency.
>
> > Q3. Our PCP-MAE is with limited generality.
>
> We demonstrate the generality of our method through demo experiments, including incorporating the proposed PCM into Point-BERT and Point-FEMAE. Without extensive exploration of architecture and the hyperparameter $\eta$, we still achieve improvement over the original methods:
>
> Methods|OBJ-BG|OBJ-ONLY|PB-T50-RS
> -|-|-|-|
> Point-BERT|87.43|88.12|83.07
> Point-BERT+PCP ($\eta=2.0$)|89.32|89.84|84.77
> Point-FEMAE $^\dagger$|94.32|92.94|89.38
> Point-FEMAE+PCP ($\eta=0.1$) $^\dagger$|95.00|93.45|89.83
>
> $^\dagger$ Reproduced results adopting our augmentation $i.e.$, Scale&Translate+Rotation for pre-training.
>
> **We believe that simply by varying $\eta$, we can achieve further improvements, and additional exploration of the architecture could lead to even higher performance. This also indicates that there is substantial room for further exploration, contradicting the notion of limited generality as you suggested.** We leave the development of a universal PCP incorporation framework to future work.

---

> ### Author Response · Authors · 2024-08-11
> **Response 3 (Response 1 continued.)**
>
> > Q4. Our PCP-MAE is with minimal inspiration.
>
> We strongly believe our PCP-MAE is well-motivated and novel. And our approach aligns closely with the found motivation. **This perspective is shared not only by us but also by four other reviewers (Reviewer MoXR, nJYR, DT8p, rD3v) who agree that our method is innovative and sheds new light on the differences in positional encodings between 2D MAE and Point-MAE.** It opens up new directions for improvement in the field of point cloud self-supervised learning.
>
> To the best of our knowledge, prior to our work, no one had observed that the decoder in 3D SSL does not necessarily rely on the encoder's representation before which is significantly different in the 2D counterpart. This led us to hypothesize that the distinction arises from the differing roles of positional encodings in 3D versus 2D contexts, prompting us to develop a straightforward yet effective approach, PCP-MAE.
>
> Point-MAE has been a foundational model in the 3D SSL landscape, with most leading methods such as ACT, ReCon, I2P-MAE, and Point-FEMAE building upon it. Enhancing this influential model carries significant importance.
>
> Our method not only highlights a potential drawback in existing MAE-based frameworks but also inspires the development of other 3D data representations, such as meshes and voxels. In these contexts, the application of positional encodings might require reevaluation, particularly if they contain context-specific information, as opposed to merely context-agnostic details.
>
> > Q5. The words that the authors want to say.
>
> Setting aside the fact that our method improves upon Point-MAE with affordable additional computational demands, achieves state-of-the-art in some benchmarks, and enhanced training efficiency compared to other SOTA methods, **it is crucial to recognize that in high-quality research, performance is not the sole criterion. The core idea and its contribution to advancing thought in the field merit greater attention.** A valuable research work should be evaluated not only by its performance metrics and quantitative outcomes but also by its potential to inspire deeper understanding and innovation within the relevant community.
>
> Thus, while you might focus on the marginal numerical improvements our PCP-MAE method offers over the much more complex cross-modal method, ReCon, we encourage you to also consider the deeper insights it provides. Exploring the true significance and advantages of our method will reveal the broader value of our work.
>
> Kind regards, The Authors

---

> ### Author Response · Authors · 2024-08-13
> **Look forward to further reply.**
>
> Dear Reviewer GmbZ:
>
> We would like to express our sincere gratitude for your thorough and insightful review comments. Your feedback has been invaluable in helping us refine and improve our paper significantly.
>
> In our most recent discussion, we have provided the **training logs** for our model. These logs offer a transparent and verifiable record of our method's performance, allowing you to easily confirm that our approach indeed achieves the results reported in the paper. This step towards greater reproducibility is part of our commitment to scientific rigor and openness.
>
> We are particularly interested in understanding whether these newly provided training logs have influenced your perspective on our work.
>
> We value your expertise and judgment, and we are eager to hear your thoughts on this new information. If you have any remaining questions or areas where you feel further clarification is needed, please don't hesitate to let us know.

---

> ### Comment · Reviewer_GmbZ · 2024-08-14
>
> Dear All,
>
> Based on the author's recommendation, I retrained PCP-MAE from scratch and conducted experiments on three variations of ScanObjectNN using ten different seeds for each. The experimental results are as follows:
>
> **Best result of ten experiments.**
>
> | Methods  |  OBJ-BG |  OBI-ONLY | PB-T50-RS  |
> | ------------ | ------------ | ------------ | ------------ |
> | Point-MAE  | 93.12  | 92.77  | 89.04  |
> | PCP-MAE （Reproduce） | 95.01  | 93.29  | 89.76  |
> | PCP-MAE （Author） | 95.53  | 94.32 | 90.35  |
> | Recon  |  95.19 | 93.12  | 90.25  |
>
> **Average result of ten experiments.**
>
> | Methods  |  OBJ-BG |  OBI-ONLY | PB-T50-RS  |
> | ------------ | ------------ | ------------ | ------------ |
> | Point-MAE  | 92.67  | 92.31  | 88.59  |
> | PCP-MAE（Reproduce）  | 94.46  | 92.65  | 89.37  |
> | PCP-MAE （Author） | 94.56  | 93.35 | 89.57  |
> | Recon  |  94.37 | 92.51  | 89.95  |
>
> I only provided the latest reproduction results. My recommendation is borderline (neither accept nor reject), and I fully respect the final decision of the AC.

---

### Official Review · Reviewer_DT8p · 2024-07-14

**Soundness:** 3
**Presentation:** 4
**Contribution:** 3
**Rating:** 4
**Confidence:** 5

**Summary:**

The paper studies the point cloud pretraining under the self-supervised learning paradigm. Authors experimentally found that the position embedding in the decoder may decrease the learning ability of the encoder and propose a new method to overcome this issue.

**Strengths:**

1. The paper is clearly written and easy to follow.
2. The phenomenon revealed in this paper is new to me and sounds reasonable.
3. The proposed method seems reasonable and aligns with the core motivation of the paper.

**Weaknesses:**

1. In line 30, authors mentioned the difficulty in collecting 3D dataset and want to use SSL to solve this problem. However, 3D data is different with 2D images which can be collected easily on the internet. For the experimented dataset like Scanning datasets, the label will be collected during scanning, then why SSL for 3D is important?
2. According to table 1, Point-FEMAE achieves almost the same performance as the proposed model, thus it is hard to say that the reconstructing objective may make the encoder unable to learn semantic features in Line 70.
3. The improvements of the proposed method over Point-FEMAE are marginal.
4. One possible reason behind the phenomenon of the position embedding may be the simplicity of the dataset. Would the phenomenon also hold for lidar datasets?

**Questions:**

pealse refer to the weakness

---

> ### Author Rebuttal · Authors · 2024-08-03
>
> Thank you for taking the time to review our paper and for providing your detailed feedback. Below, we answer your questions in detail:
>
> > Q1. The importance of SSL for 3D.
>
> Upon revisiting our statement, we find it more appropriate to remove "collect" in line 27 of the original paper, which we will update in the next version. Here are clarifications regarding your question.
>
> We agree with you that collecting 3D data is easy through scanning. However, to our knowledge, most labeled 3D datasets, such as ScanNet [1] and NuScenes [2], which contain vast amounts of 3D data, require laborious annotation, and labels cannot be directly assigned or collected within them.
>
> Take autonomous driving as an example, a data acquisition vehicle can collect more than 200k frames of point clouds within 8 working hours, but a skilled worker can only annotate 100-200 frames per day [3]. Thus, effectively leveraging unlabeled 3D data becomes a critical issue in practical applications. **Self-supervised learning in 3D can effectively leverage large amounts of unlabeled 3D data by designing specific pre-text tasks for pre-training, benefiting downstream tasks.** Therefore, 3D SSL is of significant importance.
>
> > Q2. According to table 1, Point-FEMAE achieves almost the same performance as the proposed model, thus it is hard to say that the reconstructing objective may make the encoder unable to learn semantic features in Line 70.
>
> Thank you for your insightful question. It's crucial to focus on the word **"may"** in our statement. We have observed that reconstruction does not necessarily depend on the encoder’s representation when positional embeddings (centers) are directly provided, which underscores the significance of these embeddings. Our proposed PCP-MAE model addresses this by guiding the encoder to learn distributions of semantically rich centers, replacing the directly provided centers.
>
> **However, apart from directly guide the encoder to enhance its understanding on the centers to address the found position leakage issue, other approaches could indirectly alleviate it.** Take Point-FEMAE as an example. It adds a branch to the pre-training model which prevents the model from learning limited representations. This will possibly enhance the model's understanding to the point cloud better and indirectly enhance its grasp of center distributions. Consequently, the position leakage issue may be alleviated. That's why we use **may** in our statement. In essence, while the reconstruction objective could impede learning semantic features, it can be alleviated with auxiliary tools.
>
> **Most importantly, our approach is the first to directly address the position leakage issue**. Although Point-FEMAE could also possibly alleviate this issue, it significantly lags behind in efficiency (check the answer to the Q3).
>
> > Q3. Marginal improvement over Point-FEMAE.
>
> Our PCP-MAE and Point-FEMAE are two orthogonal works based on Point-MAE. Point-FEMAE proposes a global branch and a local branch to capture latent semantic features, rather than using just one branch. Our PCP-MAE, on the other hand, guides the model to learn to predict centers based on a motivating observation.
>
> The reported results show that both PCP-MAE and Point-FEMAE significantly improve the baseline performance, with our PCP-MAE marginally outperforming Point-FEMAE.
>
> **However, when it comes to training efficiency, Point-FEMAE significantly lags behind our approach.** Point-FEMAE not only adds an additional branch but also uses extra Local Enhancement Modules for modeling the local point clouds. This results in a significant increase in both parameter and time costs compared to Point-MAE. The comparisons among these three methods are shown as follows:
>
> Method|Pre-training||Fine-tuning|||||
> -|-|-|-|-|-|-|-
> ||Params (M)|Time (s/epoch)|Params (M)|OBJ-BG Time (s/epoch)|OBJ-ONLY Time (s/epoch)|PB-T50-RS Time (s/epoch)|ModelNet40 Time (s/epoch)
> Point-MAE (baseline)|29.0|88|22.1|10|10|49|29
> Point-FEMAE|41.5 (1.43x)|326 (3.70x)|27.4 (1.24x)|30 (3.00x)|30 (3.00x)|148 (3.02x)|72 (2.48x)
> PCP-MAE (ours)|29.5 (**1.01x**)|120 (**1.36x**)|22.1 (**1.00x**)|10 (**1.00x**)|10 (**1.00x**)|49 (**1.00x**)|30 (**1.01x**)
>
>
> Point-FEMAE retains the Local Enhancement Modules during fine-tuning, which adds an unignorable extra computational burden and thus greatly decreases the speed of model fine-tuning.
>
> > Q4. One possible reason behind the phenomenon of the position embedding may be the simplicity of the dataset. Would the phenomenon also hold for lidar datasets?
>
> Thank you for your nice suggestion. In addition to the pre-training ShapeNet dataset, we have experimented with setting the mask ratio to 100% on another dataset called ScanObjectNN [6]. This is a much more challenging dataset, sampled from real-world scans that include background and occlusions, and is built based on two popular real scene mesh datasets, SceneNN [7] and ScanNet [1]. **When only using the decoder (mask 100%) for pre-training, the point cloud still can be reconstructed well through experimenting.**
>
> From a statistical perspective, the Chamfer Distance L2 Loss (Eq. 15) is even lower than that of the simpler ShapeNet dataset. The reconstruction losses are as follows:
>
> |Dataset|Epoch 1 Loss|Epoch Last Loss|
> |-|-|-|
> |ShapeNet|0.15823|0.00324|
> |ScanObjectNN|0.44190|0.00250|
>
> We will add the visualizations in the next version of our paper. Additionally, this phenomenon is likely to hold for lidar datasets, which we will test and update the results in the next version.
>
> Please check the official comment for the Reference list.

---

> ### Author Response · Authors · 2024-08-06
>
> Note that the pre-training time cost contradicts the statistics in our paper's Table 1, which is reported from Point-FEMAE. It seems Point-FEMAE provides incorrect time efficiency comparison results, which we correct here and will update in the next version of our main paper.
>
> To ensure a fair time comparison, the code for Point-MAE should be modified slightly in two ways:
>
> 1. Add "config.dataset.train.others.whole = True" to the training to align Point-FEMAE and our method.
> 2. Instead of using KNN_CUDA, change it into the knn_point function (refer to the official code of ReCon [4] or Point-FEMAE [5]) which directly uses torch operation to align with Point-FEMAE and our approach. This will significantly increase the training speed.
>
> Reference
>
> [1] Dai A, Chang A X, Savva M, et al. Scannet: Richly-annotated 3d reconstructions of indoor scenes[C], CVPR 2017.
>
> [2] Caesar H, Bankiti V, Lang A H, et al. nuscenes: A multimodal dataset for autonomous driving[C], CVPR 2020.
>
> [3] Mao J, Niu M, Jiang C, et al. One million scenes for autonomous driving: Once dataset[J]. arXiv preprint arXiv:2106.11037, 2021.
>
> [4] Qi Z, Dong R, Fan G, et al. Contrast with reconstruct: Contrastive 3d representation learning guided by generative pretraining[C], ICML 2023.
>
> [5] Zha Y, Ji H, Li J, et al. Towards compact 3d representations via point feature enhancement masked autoencoders[C], AAAI 2024.
>
> [6] Uy M A, Pham Q H, Hua B S, et al. Revisiting point cloud classification: A new benchmark dataset and classification model on real-world data[C], ICCV 2019.
>
> [7] Hua B S, Pham Q H, Nguyen D T, et al. Scenenn: A scene meshes dataset with annotations[C]//2016 fourth international conference on 3D vision (3DV). Ieee, 2016: 92-101.

---

> ### Author Response · Authors · 2024-08-13
> **Look forward to your further reply.**
>
> Dear Reviewer DT8p:
>
> Thanks again for the valuable feedback and constructive suggestions. As the discussion phase is coming to an end, we have provided our understanding and perspective on the significance of self-supervised learning for point clouds in our rebuttal, along with new experimental results. We would like to know if our rebuttal has addressed your concerns, and if you have any further questions that need clarification from us.

---

### Official Review · Reviewer_nJYR · 2024-07-16

**Soundness:** 3
**Presentation:** 2
**Contribution:** 2
**Rating:** 6
**Confidence:** 3

**Summary:**

This work examines representation learning of 3D point clouds using masked autoencoding. A known issue with such an approach is that the coordinates of the patch centers leak significant information about the geometry and semantics of the shape being reconstructed, which degrades the representations learned by these approaches. This work proposes to add an extra criterion, whereby the model predicts the positions of the masked patch centers. Pretraining is performed on ShapeNet. This simple change leads to significant improvements over Point-MAE when finetuning on ScanObjectNN and ModelNet40 for 3D classification, 3d scene segmentation, and object part segmentation.

**Strengths:**

**Originality**
* The proposed point center prediction objective is novel to the best of my knowledge, and requires several subtle innovations on the implementation side to yield improved performance (e.g., choice of sin-cos positional embeddings before MLP projection module, re-using predicting positions in the reconstruction objective with a stop-gradient operation, and the shared network architecture with cross-attention for point center prediction). Ablation experiments demonstrate the importance of each of these factors.

**Clarity**
* The work is reasonably clear, enough that the reader can understand basic ideas, motivation, and details of the proposed method. Releasing code with the paper will greatly improve reproducibility and impact.

**Quality**
* The work is of a reasonable quality; experiments and numerical results on standard benchmarks and standard evaluation setups.

**Significance**
* This work contributes to a large body of work on MAE-based representation learning for 3D point clouds, and provides an interesting and simple approach for addressing a known limitation of these approaches. Improvements over Point-MAE, the most similar baseline, are notable.

**Weaknesses:**

* Improvements over previous MAE-based methods, such as Point-FEMAE, are quite marginal, and perhaps not statistically significant. While the contributions of other MAE related methods are orthogonal and could be potentially combined with the proposed method, it is unclear if the issue of 3D center point position leakage is still significant when combined with other tools.
* Numerical results are primarily restricted to end-to-end fine-tuning. Would be curious to see the results of linear or frozen evaluations to assess the quality of the representations, as finetuning from scratch (i.e., random representations) can already achieve much of the topline performance in many cases.

**Questions:**

* If the authors have the capacity to conduct such an experiment, I am primarily curious to see the effect of the proposed center point prediction task with stronger baselines which provide advances orthogonal to the contributions in this work.

**Limitations:**

N.A.

---

> ### Author Rebuttal · Authors · 2024-08-02
>
> Thank you for the time, thorough comments, and nice suggestions. Your endorsement of our method and experiments gives us significant encouragement. Here are our clarifications.
>
> > Q1.1. Improvements over previous MAE-based methods, such as Point-FEMAE, are quite marginal, and perhaps not statistically significant.
>
> Although our PCP-MAE only marginally outperforms other advanced MAE-based methods such as Point-FEMAE and ReCon, **our approach significantly advances in efficiency,** particularly regarding pre-training and fine-tuning time costs. The efficiency comparisons are shown below (experiments conducted using an empty RTX 3090):
>
> Method|Pre-training|| Fine-tuning| | | | |
> -|-|-|-|-|-|-|-
> ||Params (M)|Time (s/epoch)|Params (M)|OBJ-BG Time (s/epoch)|OBJ-ONLY Time (s/epoch)|PB-T50-RS Time (s/epoch)|ModelNet40 Time (s/epoch)
> Point-MAE (baseline)|29.0|88|22.1|10|10|49|29
> Point-FEMAE|41.5 (1.43x)|326 (3.70x)|27.4 (1.24x)|30 (3.00x)|30 (3.00x)|148 (3.02x)|72 (2.48x)
> ReCon|140.9 (4.85x)|452 (5.14x)|43.6 (1.97x)|13 (1.30x)|13 (1.30x)|60 (1.22x)|42 (1.45x)
> PCP-MAE (ours)|29.5 (**1.01x**)|120 (**1.36x**)|22.1 (**1.00x**)|10 (**1.00x**)|10 (**1.00x**)|49 (**1.00x**)|30 (**1.01x**)
>
> > Q1.2. While the contributions of other MAE-related methods are orthogonal and could be potentially combined with the proposed method, it is unclear if the issue of 3D center point position leakage is still significant when combined with other tools.
>
> Thank you for your good question. First, 3D center point position leakage results from the Point-MAE or exactly the reconstruction objective due to the directly provided centers. Point-MAE remains the core component of MAE-based methods such as Point-FEMAE and ReCon, and point cloud reconstruction continues to be one of the pre-training objectives. **Thus, the issue of position leakage still exists in these methods.** Our approach directly guides the encoder to learn distributions of centers, which helps to mitigate this issue.
>
> **However, the extent of the negative impact caused by position leakage varies in different MAE-related methods.** Take Point-FEMAE [2] as an example. It adds a branch to the pre-training model which prevents the model from learning limited representations. This will possibly enhance the model's understanding to the point cloud better and indirectly enhance its grasp of center distributions. Consequently, the position leakage issue may be alleviated. Although the performance of these methods can reveal the degree of influence posed by position leakage, a quantitative method to analyze this issue is necessary and could be addressed in our future research.
>
> > Q2. Would be curious to see the results of linear or frozen evaluations to assess the quality of the representations.
>
> Following ReCon [1], we assess the performance of MLP-Linear, which freezes the pre-trained encoder and only updates the classification head to evaluate the quality of the learned representations. Here are the experiment results:
>
> MLP-Linear|OBJ-BG|OBJ-ONLY|PB-T50-RS|
> -|-|-|-
> Point-MAE|82.77±0.30|83.23±0.16|74.13±0.21
> ACT (Cross-modal)|85.20±0.83|85.84±0.15|76.31±0.26
> Point-FEMAE|88.98±0.15|89.50±0.22|80.32±0.10
> ReCon (Cross-modal)|89.50±0.20|89.72±0.17|81.36±0.14
> PCP-MAE (ours)|89.41±0.13|89.41±0.26|80.63±0.08|
>
> The results show that our approach greatly outperforms the baseline Point-MAE, marginally surpasses Point-FEMAE, and lags behind the leading cross-modal method ReCon.
>
> > Q3. The effect of the proposed center point prediction task with stronger baselines which provide advanced orthogonal improvement to the contributions in this work.
>
> Thank you for this nice suggestion. We tried to incorporate our center point prediction task with Masked Point Modeling (MPM) family methods include Point-BERT [3] and Point-FEMAE [2] among which Point-BERT is parallel to the Point-MAE [4] and Point-FEMAE provide advanced orthogonal improvement based on Point-MAE compared to this work.
>
> Although intuitively feasible, it is important to note that implementing our method into various MPM-based frameworks requires careful consideration due to the different optimization goals of each specific approach. **We incorporated PCP (learning predicting masked centers) into MPM methods including Point-BERT and Point-FEMAE with additional minor changes to maintain architecture consistency.** The experiment results show that incorporating our center point predicting task enhances performance when other experiment setups are aligned:
>
> Methods|OBJ-BG|OBJ-ONLY|PB-T50-RS
> -|-|-|-|
> Point-BERT|87.43|88.12|83.07
> Point-BERT+PCP ($\eta=2.0$)|89.32|89.84|84.77|
> Point-FEMAE $^\dagger$|94.32|92.94|89.38
> Point-FEMAE+PCP ($\eta=0.1$) $^\dagger$|95.00|93.45|89.83|
>
> $^\dagger$ Reproduced results adopting our augmentation $i.e.$, Scale&Translate+Rotation for pre-training.
>
> Due to time constraints, exploration of various design choices and different $\eta$ was not feasible. We believe that further customization could lead to more substantial improvements.
>
> [1] Qi Z, Dong R, Fan G, et al. Contrast with reconstruct: Contrastive 3d representation learning guided by generative pretraining[C], ICML 2023.
>
> [2] Zha Y, Ji H, Li J, et al. Towards compact 3d representations via point feature enhancement masked autoencoders[C], AAAI 2024.
>
> [3] Yu X, Tang L, Rao Y, et al. Point-bert: Pre-training 3d point cloud transformers with masked point modeling[C], CVPR 2022.
>
> [4] Pang Y, Wang W, Tay F E H, et al. Masked autoencoders for point cloud self-supervised learning[C], ECCV 2022.

---

### Official Review · Reviewer_MoXR · 2024-07-16

**Soundness:** 3
**Presentation:** 2
**Contribution:** 3
**Rating:** 5
**Confidence:** 4

**Summary:**

The paper proposes a masked autoencoding based self-supervised approach for 3D point clouds. The approach which terms as PCP-MAE, learns to predict centers for Point Masked AutoEncoders. The paper investigate that the coordinates of the centers are essential in the point cloud field and the decoder can even reconstruct the masked patches without the encoder representation of the visible patches. Thus, the paper introduces another pre-training objective which predicts the centers and uses them to replace directly provided centers, leading to improved performance. Moreover, this allows the network to not only encode visible centers effectively but also learn the inter-relationships between visible and masked centers. Finally, the approach is efficient and outperforms Point-MAE on multiple benchmark datasets.

**Strengths:**

- Provide an insight on why only providing center is good enough for point cloud reconstruction in masked autoencoding.
- Propose an additional objective task that learns to predict the center of the masked patches and the use of predicted center of the masked patches in the decoder instead of ground-truth center.
- Experiments results on ScanObjectNN, ModelNet40 and S3DIS.
- Extensive ablation studies on different components in PCP-MAE, loss functions, masking ratio and other architecture related modules.

**Weaknesses:**

- The paper writing can definitely be improved and sometimes it's really to hard to follow the underlying idea behind it.
- The figure 2 can be misleading and it doesn't correlate with the equation 10. In figure 2, it looks like only the masked tokens are passed to the PCP module but in equation 10 which is given as:$PE_{m}^{pred} = PCM (E_{m}, E_{v}, PE_{v})$, it looks like PCP module uses encoded representation of masked patches, visible patches and positional embeddings of visible patches. Can the authors please clarify about this?
- In Table 2, PointMAE achieves 90.02 88.29 85.18 on three variants of ScanObjNN but in the ablation study (c.f. Table 5 ) the results on the 4th row ( just reconstruction only ) achieves 92.94 92.42 88.65. Isn't only doing reconstruction equivalent to PointMAE? Does it mean the performance boost comes from the positional encoding?
- Although approach looks simpler and effective, but it only shows improvement by incorporating the additional details on PointMAE. Compared to other the baselines like Point-FEMAE, it doesn't outperforms them on multiple datasets. Can the authors please provide additional results by incorporating PCP and new positional encoding in multiple MAE approaches for point clouds?
- Did the authors adapt the same set of augmentations for pre-training baselines?
- The methods looks very sensitive to $\eta$ (c.f. Figure 4)
- Some additional results on indoor datasets like ScanNet would be helpful.

**Questions:**

Please see the questions and the suggestions in the weakness section.

**Limitations:**

Yes, the authors have adequately addressed the limitations.

---

> ### Author Rebuttal · Authors · 2024-08-02
>
> Thank you for the time, thorough comments, and nice suggestions. We are pleased to clarify your questions one-by-one.
> > Q1. The paper writing should be improved.
>
> Thank you for your feedback. We will improve the manuscript's language and structure for better clarity and readability, and these updates will be reflected in the next version.
> > Q2. The figure 2 can be misleading and it doesn't correlate with the equation 10.
>
> Thank you for pointing out this misleading detail. We have changed Figure 2 to align with Equation 10. Please check the attached rebuttal PDF.
> > Q3. Isn't only doing reconstruction equivalent to PointMAE? Does it mean the performance boost comes from the positional encoding?
>
> Yes and no. They are equivalent except for a minor change in the positional encoding. We experimented on the effect of this behavior when aligning other setups:
> Method|OBJ-BG|OBJ-ONLY|PB-T50-RS
> -|-|-|-
> Point-MAE|92.94|92.25|88.86
> Reconstruction_only|92.94|92.42|88.65
>
> The results show that it only slightly influences performance.
>
> The performance boost mainly comes from the augmentations, i.e., the rotation augmentation explored by [1] for fine-tuning and the Scale & Translate + Rotation pre-training augmentation explored by us. The ablation results of Point-MAE are shown as follows:
> Augmentation||OBJ-BG|OBJ-ONLY|PB-T50-RS
> -|-|-|-|-
> Pre-training|Fine-tuning|
> Scale&Translate|Scale&Translate $^*$ |90.02|88.29|85.18
> Rotation|Rotation $^\dagger$|92.60|91.91|88.42|
> Scale&Translate+Rotation|Rotation $^\ddagger$|92.94|92.25|88.86
>
> $^*$ Adopted by Point-MAE.
>
> $^\dagger$ Adopted by other SOTA methods including Point-FEMAE and ReCon.
>
> $^\ddagger$ Adopted by our PCP-MAE.
>
> Note that Scale&Translate+Rotation pre-training augmentation benefits both Point-MAE and our PCP-MAE but does not benefit other SOTA methods, including Point-FEMAE and ReCon (experiment results are in Q5).
> > Q4.1. Compared to other the baselines like Point-FEMAE, it doesn't outperforms them on multiple datasets.
>
> Although our PCP-MAE only marginally outperforms previous SOTA methods like Point-FEMAE and ReCon or performs comparably, **our PCP-MAE significantly exceeds them in efficiency.** The comparisons among these three methods are shown below (using one RTX 3090):
>
> Method|Pre-training||Fine-tuning|||||
> -|-|-|-|-|-|-|-
> ||Params (M)|Time (s/epoch)|Params (M)|OBJ-BG Time (s/epoch)|OBJ-ONLY Time (s/epoch)|PB-T50-RS Time (s/epoch)|ModelNet40 Time (s/epoch)
> Point-MAE (baseline)|29.0|88|22.1|10|10|49|29
> Point-FEMAE|41.5 (1.43x)|326 (3.70x)|27.4 (1.24x)|30 (3.00x)|30 (3.00x)|148 (3.02x)|72 (2.48x)
> ReCon|140.9 (4.85x)|452 (5.14x)|43.6 (1.97x)|13 (1.30x)|13 (1.30x)|60 (1.22x)|42 (1.45x)
> PCP-MAE (ours)|29.5 (**1.01x**)|120 (**1.36x**)|22.1 (**1.00x**)|10 (**1.00x**)|10 (**1.00x**)|49 (**1.00x**)|30 (**1.01x**)
> > Q4.2. Additional results by incorporating PCP and new positional encoding in multiple MAE approaches
>
> Although intuitively feasible, it is important to note that implementing our method into various MAE-based frameworks requires careful consideration due to the different optimization goals of each specific approach.
>
> **We incorporated PCP into MPM (Masked Point Modeling) methods including Point-BERT [4] and Point-FEMAE [5] with additional minor changes to maintain architecture consistency.** The results demonstrate that PCP enhances performance when other setups are aligned:
> Methods|OBJ-BG|OBJ-ONLY|PB-T50-RS
> -|-|-|-|
> Point-BERT|87.43|88.12|83.07
> Point-BERT+PCP ($\eta=2.0$)|89.32|89.84|84.77
> Point-FEMAE $^\dagger$|94.32|92.94|89.38
> Point-FEMAE+PCP ($\eta=0.1$) $^\dagger$|95.00|93.45|89.83
>
> $^\dagger$ Reproduced results adopting our augmentation $i.e.$, Scale&Translate+Rotation for pre-training.
>
> Due to time limits, we couldn't explore different design choices and $\eta$, which could bring improvement.
> >Q5. Did the authors adopt the same set of augmentations for pre-training baselines?
>
> We used the Scale&Translate+Rotation augmentation for pre-training because we found that this combination benefits our PCP-MAE and slightly benefits Point-MAE, **which is different from previous baselines (Tab. 10).** This improvement can be attributed to the fact that the combination of augmentations enables a variety of centers, allowing our PCP-MAE to learn a richer distribution of center information and become more robust. **In contrast, previous SOTA methods like ReCon and Point-FEMAE do not benefit from this augmentation empirically.**
>
> The detailed augmentations are illustrated in Q3. We further provide the results after aligning the augmentations of Point-FEMAE and ReCon with ours for clarity:
> Method|OBJ-BG|OBJ-ONLY|PB-T50-RS
> -|-|-|-
> Point-FEMAE (Origin)|95.18|93.29|90.22
> Point-FEMAE (Our augmentation)|94.32|92.94|89.38
> ReCon (Origin)|95.18|93.63|90.63
> ReCon (Our augmentation)|94.49|92.77|89.55
> >Q6. The methods looks very sensitive to $\eta$ (Figure 4).
>
> We tried values from 0.1 to 1.0 in increments of 0.1 for convenience and the results in Figure 4 show a small $\eta$ is suitable for our PCP-MAE. When choosing more fine-grained $\eta$, our approach behaves less sensitively:
>
> $\eta$|0.02|0.04|0.06|0.08|0.10|0.12|0.14|0.16|0.18
> -|-|-|-|-|-|-|-|-|-
> ScanObjectNN (OBJ-BG)|94.14|94.66|94.49|95.35|**95.53**|95.00|94.66|94.49|94.66
> ModelNet40 (w/o voting)|93.56|93.48|93.64|93.72|**93.96**|93.84|93.52|93.44|93.52
> >Q7. Some additional results on indoor datasets like ScanNet would be helpful.
>
> We experimented with the indoor dataset S3DIS [2] (lines 275-279), which provides detailed indoor 3D point cloud of over 6,000 square meters across six buildings. The scene segmentation results are:
> Methods|mAcc|mIoU
> -|-|-
> Point-MAE|69.9|60.8
> PCP-MAE (ours)|71.0|61.3
>
> For ScanNet [3], the dataset's different statistics (e.g., 2k input points for ShapeNet vs. 50k for ScanNet) require changes to the model architecture and pre-training setups. Due to time and resource constraints, we couldn't complete this but will update the results in the next paper version.

---

> > ### Comment · Reviewer_MoXR · 2024-08-12
> > **Response to Authors**
> >
> > Thanks for the detailed rebuttal. The authors have addressed some of my concerns regarding the paper diagram and promise to improve the manuscript. I have also read other reviewers comments and rebuttals by the authors and I conclude that although the PCP-MAE shows some improvement compared to other SOTA methods, I agree with Reviewer GmbZ that most the improvement comes from the other tricks like augmentation. Moreover, I would encourage the authors to study the impact of proposed approach in other methods more rigorously. Furthermore, I understand the randomness in current 3D datasets, so it would be important to study the impact of the approaches on datasets like ScanNet. Even on S3DIS dataset, the approach doesn't show any improvement compared to ReCon, Point-FEMAE. So, I will maintain my original score.

---

> > > ### Author Response · Authors · 2024-08-13
> > > **Clarification on Method's Effectiveness: Beyond Augmentation Tricks**
> > >
> > > Thank you for your response, which gives us the opportunity to clarify that the benefits of our method do not stem from augmentation tricks.
> > >
> > > Firstly, we attempted to apply the mentioned augmentation tricks to previous SOTA methods, but this did not yield improvements and even showed a declining trend, as demonstrated below:
> > >
> > > Method|OBJ-BG|OBJ-ONLY|PB-T50-RS
> > > -|-|-|-
> > > Point-FEMAE (+ Our augmentation)|94.32|92.94|89.38
> > > ReCon (+Our augmentation)|94.49|92.77|89.55
> > > PCP-MAE (Ours) |  **95.52** | **94.32** | **90.35**
> > >
> > > Besides, we believe that different methods are suited to **different hyperparameters** (including augmentation), and it is widely accepted to apply new hyperparameters to the newly proposed methods. Moreover, we have also provided results for the baseline method under **our hyperparameter** settings for **fair comparison**, which show that our method significantly outperforms both the baseline and other state-of-the-art methods:
> > >
> > > Methods|OBJ-BG|OBJ-ONLY|PB-T50-RS
> > > -|-|-|-
> > > Point-MAE (baseline) |92.60|91.91|88.42
> > > Point-MAE (+ our aug) |92.94|92.25|88.86
> > > Ours |  **95.52** | **94.32** | **90.35**
> > >
> > > Furthermore, our method (PCP) can serve as a **plug-in** module, applicable to a wide range of models. This demonstrates that our approach is not merely a trick, but a method with broad applicability:
> > >
> > > Methods|OBJ-BG|BJ-ONLY|PB-T50-RS|
> > > -|-|-|-
> > > Point-BERT|87.43|88.12|83.07
> > > Point-BERT+PCP ($\eta$=2.0)|89.32|89.84|84.77|
> > > Point-FEMAE + (Our Aug)|94.32|92.94|89.38
> > > Point-FEMAE+PCP ($\eta$=0.1) + (Our Aug)| **95.00** | **93.45** | **89.83**
> > >
> > > Additionally, we would like to reiterate that one of our contributions is revealing a vulnerability in the Point MPM (Masked Point Modeling) series of methods. Specifically, we showed that it is possible to reconstruct masked point clouds without needing an encoder or decoder. This challenges the assumptions of many researchers and can stimulate deeper reflection in the field.

---

> > > > ### Comment · Reviewer_MoXR · 2024-08-13
> > > > **Thanks for the clarifications**
> > > >
> > > > I appreciate the authors for further clarifications on the approach and the experiments. However, I'm still not fully convinced on the results. For example if we remove the augmentation trick from the approach, the method doesn't outperform ReCon, Point-FEMAE on ScanOBJNN (c.f. Table 10, row 3). As I mentioned before, given the randomness in 3D datasets, I'm more interested in the comparison on other datasets, even on S3DIS, I don't see a significant difference in the results (c.f Table 4).

---

> ### Author Response · Authors · 2024-08-07
>
> Reference
>
> [1] Dong R, Qi Z, Zhang L, et al. Autoencoders as cross-modal teachers: Can pretrained 2d image transformers help 3d representation learning?[J]. arXiv preprint arXiv:2212.08320, 2022.
>
> [2] Armeni I, Sener O, Zamir A R, et al. 3d semantic parsing of large-scale indoor spaces[C], CVPR 2016.
>
> [3] Dai A, Chang A X, Savva M, et al. Scannet: Richly-annotated 3d reconstructions of indoor scenes[C], CVPR 2017.
>
> [4] Yu X, Tang L, Rao Y, et al. Point-bert: Pre-training 3d point cloud transformers with masked point modeling[C], CVPR 2022.
>
> [5] Zha Y, Ji H, Li J, et al. Towards compact 3d representations via point feature enhancement masked autoencoders[C], AAAI 2024.

---

### Author Rebuttal · Authors · 2024-08-07

We sincerely thank the reviewers for their detailed and valuable suggestions and are grateful for the encouraging comments:

1. The paper is clearly written and easy to follow. (DT8p, nJYR, rD3v)
2. The core observation and motivation remain novel and clear. This paper is well-motivated and provides new insights into the differences in positional encodings between 2D MAE and Point-MAE. It opens up new directions for improvement in the field of point cloud self-supervised learning. (MoXR, nJYR, DT8p, rD3v)
3. The proposed method aligns well with the motivation. It is a simple and effective approach that successfully addresses the identified limitations. (MoXR, nJYR, DT8p, rD3v)
4. The experimental results sufficiently demonstrate the effectiveness of the proposed method. (MoXR, nJYR, rD3v)
5. Extensive ablation studies show the effectiveness of different components and designs. (MoXR, nJYR)

Based on the comments, we conclude some noteworthy replies for the reviewers including:

- **[Inappropriate details in the overview figure (Fig. 2). Reviewer MoXR, GmbZ]** We have clarified the details for the reviewers and modified the figure according to the feedback. Please check the attached rebuttal PDF.

- **[Marginal improvement over other SOTA methods. Reviewer MoXR, nJYR, DT8p]** In response, we highlighted the simplicity and efficiency of our method. Compared to other SOTA methods, we require significantly less training time and fewer parameters. Specifically, Point-FEMAE requires 326s per epoch, whereas our PCP-MAE only necessitates 120s per epoch. Our approach is 2.7x faster than the Point-FEMAE.

- **[The generality of our approach. Reviewer MoXR, nJYR, GmbZ]** We have incorporated our core idea into different methods, including Point-BERT and Point-FEMAE. Both demonstrate performance gains after integrating our idea, showcasing the great generality of our approach.

- **[Seemingly unfair comparison and concerns about the effectiveness of our approach. Reviewer GmbZ]** Given that Reviewer GmbZ tested our idea personally, we kindly remind them that there may be some subtle design aspects that should be carefully implemented. The reviewer also believes that all our improvements stem from the different pre-training augmentations rather than the method itself, leading to potentially unfair comparisons with previous SOTA methods. **To address this, our experiments show that other SOTA methods suffer performance drops when using the same augmentation, which we will add to the next version of our paper. Our method benefits from this due to its specific features, which we explain in the rebuttal.** We thank him for reminding us to display the performance of previous methods under the same settings. More importantly, we have confirmed through experiments that the effectiveness of our approach stems from the proposed method itself, rather than from the "data augmentation trick", as demonstrated by testing other methods with the same augmentation. To further enhance the reliability of our approach, we have provided the code and corresponding checkpoints to the ACs via an anonymous link.

- **[Additional experiments. Reviewer MoXR, DT8p]** We have added new experiments based on the comments, including frozen evaluations and validation of the core phenomenon on other datasets.

We sincerely hope that this work provides valuable insights into the field of point cloud self-supervised learning and that the misunderstandings raised by Reviewer GmbZ can be resolved through this rebuttal. Thanks again to all reviewers for their valuable time to help improve our work.

---

### Author Response · Authors · 2024-08-07
**Code submission**

To address the misunderstandings about the implementation and effectiveness of our approach on the part of Reviewer GmbZ, we provide the anonymous link to our PCP-MAE code to the ACs for clarification. The link is: https://anonymous.4open.science/r/2128-PCP-MAE-529D

---

### Decision · Program_Chairs · 2024-09-25

**Decision:**

Accept (spotlight)

**Comment:**

**Summary of the paper:**
The paper identifies a central problem with PointMAE, that is, the position of patch centers that are provided to the decoder leaks information about the 3D structure of the object and therefore a reconstruction of the point cloud is possible without considering any output of the encoder. Then, it suggests an updated mechanism where the encoder also predicts the center of the patches which is then used in the decoder to better encourage the encoder to embed relevant information for the reconstruction. The modified PointMAE is shown to consistently perform better at downstream tasks.

**Summary of the reviews:**
The reviewers find the insight on the different role of patch center embedding for point-clouds and fixed grids interesting, informative and important. They also find the proposed modification relevant. They finally appreciate the improved results over PointMAE and the extensive side/ablation studies.

On the other hand, the reviewers believe the presentation, text, figures, and equations can be improved, and that some of the reported results are confusing. They also asked for the addition of PCP on more recent variants of masking-based SSL on point clouds.

**Summary of the rebuttal and discussions:**
The authors clarified some confusions about the presentation and the reported numbers. They also argue for the computational efficiency of PCP-MAE over more recent state of the art methods. They further provided additional experiments that (1) show improvements when PCP is added to Point-BERT and Point-FEMAE, and (2) evaluates the method on a new indoor dataset.

Some reviewers remained unconvinced regarding the comparison to the state of the art and considered most improvements coming from the different augmentations. The authors provided their code anonymously and reviewer GmbZ put a laudable effort in reproducing results, using their code, and with 10 random seeds that makes it quite reliable.

**Consolidation report:**
Most concerns were satisfactorily refuted. The main outstanding concern is the comparison with state-of-the-art which only makes PCP-MAE comparable to them and not superior. The reproduced results of the reviewer, while slightly lower than the reported ones by the authors, show similar trends, i.e. improved performance over pointMAE but only comparable performance to the state of the art. The authors demonstrate that their method has significantly better computational efficiency.

AC’s additional take: the family of PointMAE is an active field of research both for improvement and application. The problem that the paper finds with PointMAE is fundamental and indisputable. The significance of this finding by itself was more or less undermined by the reviewers. The proposed approach, although can be improved, is simple and sensible and achieves some improvement over the main baseline and is comparable to the state of the art while being significantly more efficient. The identification of a central problem and proposing simple mitigation are considered significant steps forward for the field.

**Recommendation:**
The mere identification of such a central and important issue makes the paper newsworthy for a considerably-sized audience working with point clouds. The improved results over pointMAE and comparable results with sota with much lower computational needs, showcase the potential improvements that working on the main finding can bring about. The AC recommends acceptance.